



# Optimal feature selection for improved ML based reconstruction of Global Terrestrial Water Storage Anomalies

Nehar Mandal[1], Prabal Das[2], and Kironmala Chanda[1,3]

[1]Department of Civil Engineering, Indian Institute of Technology (Indian School of Mines) Dhanbad, India
[2]Department of Civil Engineering The University of Texas, Arlington, TX, USA, 76019
[3]Centre for Water Resources Management, Indian Institute of Technology (Indian School of Mines) Dhanbad, India

**Correspondence:** Kironmala Chanda (kironmala@iitism.ac.in)

**Abstract.** Understanding long-term Terrestrial water storage (TWS) variations is vital for investigating hydrological extreme events, managing water resources, and assessing climate change impacts. However, the limited data duration from the Gravity Recovery and Climate Experiment (GRACE) and its follow-on missions (GRACE-FO) poses challenges for comprehensive long-term analysis. In this study, we reconstruct TWS anomalies (TWSA) for the period Jan 1960 to Dec 2022 thereby filling data gaps between GRACE and GRACE-FO missions as well as generating a complete dataset for the pre-GRACE era. The workflow involves identifying optimal predictors from land surface model (LSM) outputs, meteorological variables, and climatic indices using a novel Bayesian Network (BN) technique for grid-based TWSA simulations. Climate indices, like the Oceanic Niño Index and Dipole Mode Index, are selected as optimal predictors for a large number of grids globally, along with TWSA from LSM outputs. The most effective machine learning (ML) algorithms among Convolutional Neural Network (CNN), Support Vector Regression (SVR), Extra Trees Regressor (ETR), and Stacking Ensemble Regression (SER) models are evaluated at each grid location to achieve optimal reproducibility. Globally, ETR performs best for most of the grids which is also noticed at the river-basin scale, particularly for the Ganga-Brahmaputra-Meghana, Godavari, Krishna, Limpopo, and Nile river basins. The simulated TWSA (BNML_TWSA) outperformed the TWSA from LSM outputs when evaluated against GRACE datasets. Improvements are particularly noted in the river basins such as Godavari, Krishna, Danube, Amazon, etc., with median values of the correlation coefficient, Nash-Sutcliffe efficiency, and RMSE for all grids in Godavari, India, being 0.927, 0.839, and 63.7 mm respectively. A comparison with TWSA reconstructed in recent studies indicates that the proposed BNML_TWSA outperforms them globally as well as for all the 11 major river basins examined. The presented dataset is published at https://doi.org/10.6084/m9.figshare.25376695 (Mandal et al., 2024) and updates will be published when needed.

## 1 Introduction

Terrestrial water storage (TWS) refers to the storage of water on or above the surface of the Earth, including hydrological elements including groundwater, soil moisture, snow, ice, and surface water (Yu et al., 2021; Yang et al., 2021). The fluctuations of TWS in both space and time have been comprehensively simulated by employing global land surface and hydrological models (Sun et al., 2021). These models may have significant biases due to inherent uncertainty and the lack of some physical processes. Which limits the application of these model outputs for long-term analysis of climate change impact assessment,





water resources management, and hydrological extreme event forecasting. The Gravity Recovery and Climate Experiment
(GRACE) mission and its successor GRACE Follow-on (GRACE-FO) have been providing unprecedented accurate measurements of terrestrial water storage anomalies (TWSA) since April 2002 (Mo et al., 2022). These TWSA observations have been
widely used in hydrological studies to assess the impacts of climate change and human activities on large-scale water balance,
droughts and floods, and groundwater storage (Mo et al., 2022; Rodell et al., 2018). Similar to other satellite observations, the
GRACE mission has data gaps such as an 11-month gap between the end of the GRACE mission in June 2017 and the start of
the GRACE-FO mission in May 2018, with an additional few months of data gap during each mission.

Several studies have undertaken the reconstruction of TWSA by establishing empirical relationships between GRACE
TWSA and associated climatic and hydrological factors including precipitation, temperature, soil moisture, and sea surface
temperature (Sun et al., 2020, 2021; Li et al., 2020). In a study by Becker et al. (2011), authors integrated spatial patterns of
TWS derived from GRACE data with long-term in-situ river level records to recreate the TWS for the Amazon river basin
from 1980 to 2008. Forootan et al. (2014) developed Autoregressive Model with Exogenous Variables (ARX), a statistical
data-driven approach for reconstruction of TWSA for West Africa, using GRACE dataset, rainfall data from TRMM, along
with sea surface temperature (SST) information spanning the Atlantic, Indian, and Pacific Oceans. Nie et al. (2016) utilized the
Global Land Data Assimilation System (GLDAS) products and GRACE-based TWSA data to reconstruct TWSA by water balance approach from 1948 to 2012 over Amazon river basin. Humphrey et al. (2017) established a statistical data-driven model,
between GRACE TWSA using deviations in both temperature and precipitation to recreate TWSA in the Amazon river basin
from 1985 to 2015. Machine learning (ML)-based algorithms have gained popularity during the past two decades and presented
new opportunities in hydrology and related fields, including reconstruction of TWSA. Long et al. (2014) made one of the first
attempts to hindcast TWSA for a period of February 1979 to September 2012 by developing an artificial neural network (ANN)
model using GRACE and other in-situ modeling data to study extreme climate events in the Yun-Gui Plateau, China. Sun et al.
(2019) used deep convolutional neural network (CNN) with three model architectures to predict the spatio-temporal variations
of TWSA over India. Ahmed et al. (2019) derived a Nonlinear Autoregressive with Exogenous Input (NARX) Model, to establish a relationship between GRACE TWS and precipitation, temperature, evapotranspiration and Normalized Difference
Vegetation Index. Authors conducted forecasting of GRACE TWS for 10 major river basins of Africa. Data-driven techniques
namely multiple linear regression (MLR), ANN, and autoregressive exogenous are used to reconstruct (January 1992 to March
2002) and predict (July 2017 to December 2018) TWSA by Li et al. (2020). MLR shows a robust performance for gridded
TWSA prediction and reconstruction over 26 global river basins. Sun et al. (2020) used a seasonal autoregressive integrated
moving average with exogenous variables models, deep neural network, and MLR, to reconstruct grid scale missing monthly
GRACE TWSA data over 60 global river basins. Jing et al. (2020) generated a GRACE-like TWSA prior to the GRACE period
back to 1979 over the Nile River basin using Random Forest and the eXtreme Gradient Boost ensemble learning algorithms.
Yu et al. (2021) used three deep learning models to hindcast the TWSA over Canada from 1972 to 2002 based on land surface
model (LSM) output as a predictor. Satish Kumar et al. (2023) reconstructed GRACE-like time series of TWSA from 1960 to
2016 across four river basins in southern India. Groundwater storage anomalies are generated from these reconstructed TWSA
and subsequently compared against data gathered from close to 2000 groundwater monitoring wells.





The ML models used in hydrological studies so far can be broadly divided into a few categories. Most ML investigations predominantly belong to a single algorithm usage category where the performance of a specific algorithm is evaluated against the baseline performance (Raghavendra and Deka, 2014; Sun et al., 2014; Mo et al., 2022; Khan and Maity, 2020). Other studies compare the performance of different ML models with the aim of finding the best algorithm, which performs well across a wide range of situations (Mandal and Chanda, 2023; Sun et al., 2021). In recent times a relatively new approach has been adopted

by many studies, wherein the optimal features are selected before applying a single and/or multiple ML algorithms to evaluate the prediction accuracy (Das and Chanda, 2020; Das et al., 2022). Using a single algorithmic approach could prove satisfactory when dealing with a compact research area or multiple study areas with similar hydroclimatic characteristics. However, it becomes insignificant for large study areas, multi-site analysis with different hydroclimatic conditions and where the relative importance of predictors can vary spatially. This may greatly affect the final performance of machine learning models. Previous

studies on grid scale reconstruction of GRACE TWSA indicate that across all global basins, there is no individual algorithm that consistently outperforms others (Sun et al., 2020, 2021; Li et al., 2020). Sun et al. (2020) found that a deep neural network model performed better than the other two data-driven methods for reconstructing TWSA over global river basins at a grid scale. Mo et al. (2022) employed Bayesian convolutional neural networks (BCNN) to reliably interpolate the TWSA data gap between GRACE and GRACE-FO globally. Deep convolutional autoencoders outperform CNN and BCNN when filling the

gaps between GRACE and GRACE-FO globally (Uz et al., 2022). The integrated convolutional neural network-based support vector machine has been found to outperform other regression models in the Congo River basin, Africa by Kalu et al. (2023).

To the best of our knowledge, the selection of optimal predictors for reconstruction of TWSA using Bayesian networks is first exercised in this study which also draws the novelty of the study. The input and target data sets are used without first performing interpolation of intermittent gaps, detrending, deseasoning, or decomposing signals, in order to prevent the

introduction of bias. We applied different ML models to each global grid in this study and selected the most appropriate model for each grid based on their performance to ensure optimal reproducibility. Furthermore, we conduct the analysis at the basin scale across 11 global river basins with varied hydroclimatic characteristics from 6 different continents: Amazon, Danube, Ganga-Brahmaputra-Meghana (GBM), Godavari, Indus, Krishna, Limpopo, Mississippi, Murray-Darling, Nile, and Zambezi. Among these rivers, the Amazon, GBM, and Mississippi exhibit humid hydrologic characteristics, while the Nile is semi-arid

and the Zambezi is semi-humid (Uz et al., 2022). Furthermore, a diverse range of basin sizes has been taken into account, spanning from the vast Mississippi basin, which covers 3220000 km$^2$, to the relatively small Krishna river basin, with an area of 258948 km$^2$. The study aims to achieve three objectives. First, it aims to specifically select the optimal predictors from a number of meaningful inputs including LSM outputs, meteorological variables, and climate indices, for each grid, utilizing the potential of Bayesian networks. Second, it aims to select a leader model for each grid from a number of ML models

including kernel-based, network-based and ensemble models based on their performance. Finally, it aims to simulate GRACE-like TWSA and reconstruct global TWSA datasets for the historical period starting from 1960 and including the data gap periods of GRACE and GRACE-FO missions. In the rest of the paper, Section 2 describes the data used and processing of data products. In Section 3 the methodological details of the feature section process, description of ML algorithms and the overall





workflow are presented. The results and discussions are presented in Section 4, followed by a summary of the conclusions in

Section 5.

## 2   Data and processing

The complete set of predictors used in this study includes TWSA data products from GLDAS LSMs, climate forcing data (precipitation and temperature), and a number of climate indices. Brief descriptions of data products used in this study and their sources are discussed in the subsections below.

### 2.1   GRACE terrestrial water storage anomaly (TWSA)

This study makes use of the Coastline Resolution Improved version of the GRACE mascon product (RL06.1Mv03) downloaded from the Jet Propulsion Laboratory (JPL-RL06) website (https://grace.jpl.nasa.gov). The JPL-RL06 has a 0.5°lat × 0.5°lon grid resolution however, it naturally symbolises 3°× 3°equal-area caps, which match the mass concentration functions (mascon) used to estimate and parameterize the monthly gravity fields globally. (Wiese et al., 2016; Watkins et al., 2015). The GRACE

datasets are provided as anomalies with respect to the 2004 to 2009 mean Terrestrial Water Storage (TWS). Two missions covered the observation period of the GRACE dataset: From April 2002 to June 2017, during the GRACE mission, and from June 2018 to the present day, under the GRACE-FO mission, with an 11-month gap between the two missions. Additionally, there are intermittent data gaps within each mission. In this study, short data gaps (1-2 months) are also filled using trained ML models, unlike some previous studies where the data from neighbouring months is used to fill in these gaps (Mo et al., 2022;

Yang et al., 2021).

### 2.2   Global land data assimilation system (GLDAS) simulated TWS

GLDAS LSM simulated TWS data from two different models, Catchment Land Surface Model (CLSM) (Li et al., 2019) and NOAH (Rodell et al., 2004) are used in this study. Both LSM data products are retrieved from the GES DISC, a NASA Goddard Earth Sciences Data and Information Services Center (https://disc.gsfc.nasa.gov). CLSM GLDAS CLSM025 v2.0 covers the

period January 1948 to December 2014, while GLDAS CLSM025 v2.2 spans February 2003 to the present, variables in these versions include TWS as an output. On the other hand, NOAH (GLDAS_NOAH025) TWS is estimated as an aggregate of soil moisture content (in all four layers, ranging in depth from 0 to 200 cm), canopy surface water, and snow water, which is available from January 1948 to the present and consists of two data versions. Spatiotemporal resolution and other details of all data products used in this study are presented in Table 1. To obtain the GLDAS TWSA, the long-term TWS mean from January

2004 to December 2009 is subtracted from the corresponding GLDAS TWS data (Sun et al., 2021). The corresponding TWSA from NOAH and CLSM are denoted by NTWSA and CTWSA, respectively. To hindcast the TWSA for the period January 1960 to March 2002, older versions (v2.0) of the GLDAS LSM data products are used.



**Table 1.** Information regarding the data products employed in this research (links for accessing these data products are provided in the Data Availability Statement section)

| Product | Source | Variables | Spatial Resolution (lat × lon), Temporal resolution | Version (Data period) |
|---|---|---|---|---|
| GRACE JPL mascon | Watkins et al. (2015); NASA/JPL (2023) | Terrestrial water storage anomaly | 0.50°×0.50°, 1 month | RL06.1Mv03 (Apr 2002 - Dec 2022) |
| GLDAS CLSM | Li et al. (2019) | Terrestrial water storage | 0.25°×0.25°, 1 day (aggregated to monthly) | GLDAS_CLSM025 v2.0 (Jan 1960 - Jan 2003) |
| | | | | GLDAS_CLSM025 v2.2 (Feb 2003 - Dec 2022) |
| GLDAS NOAH | Rodell et al. (2004) | Canopy surface water, Soil moisture content, Snow water, Precipitation, Temperature | 0.25°×0.25°, 1 month | GLDAS_NOAH025 v2.0 (Jan 1960 - Mar 2002) |
| | | | | GLDAS_NOAH025 v2.1 (Apr 2002 - Dec 2022) |
| Dipole Mode Index | Saji and Yamagata (2003); Saji et al. (1999) | - | 1 month | - |
| North Atlantic Oscillation index | Wallace and Gutzler (1981); Barnston and Livezey (1987) | - | 1 month | - |
| Oceanic Niño Index | Barnston et al. (1997) | - | 1 month | - |





## 2.3 Meteorological data

Meteorological data such as precipitation and temperature have been included as predictors to enhance the model's predictive
capability. Although precipitation and temperature are components of LSM forcing, the LSM doesn't utilise all of the data
in the forcing to its full potential. The amount of precipitation affects the recharge of groundwater and surface waters, while
temperature is an indicator of energy available for evapotranspiration. Hence, precipitation and temperature may capture some
specific aspect that the LSM models may not simulate accurately (Sun et al., 2019; Humphrey et al., 2017). These climate
forcing data (precipitation and temperature) are obtained from GLDAS NOAH LSM output for the period of analysis. These
products are selected because of their global coverage and successful usage in earlier investigations (Sun et al., 2019).

## 2.4 Climate Indices

Dipole Mode Index (DMI), North Atlantic Oscillation (NAO), and Oceanic Niño Index (ONI) have been widely utilized as
optimal teleconnection predictors for the seasonality of surface temperature and precipitation (Harou et al., 2006; Brandimarte
et al., 2011; Hafez et al., 2016). DMI is the anomalous SST gradient between the western and south eastern equatorial Indian
Oceans associated with the Indian Ocean Dipole (Saji et al., 1999; Saji and Yamagata, 2003). The NAO index characterises the
changes in strength between the subtropical high and subpolar low pressure patterns in the atmosphere over the North Atlantic
Ocean (Wallace and Gutzler, 1981; Barnston and Livezey, 1987). The difference between rolling three months' average sea
surface temperature in the east-central tropical Pacific and the long-term average of the same three months is characterised as
the ONI (Barnston et al., 1997). Recent studies have utilized some climate indices as predictors of TWSA (Forootan et al.,
2019; Phillips et al., 2012; Sun et al., 2021). The prediction skills are spatially impacted by specific climate phenomena or
conditions, such as El Niño and La Niña events, represented by those climate indices (Sun et al., 2021).

## 3 Methodology

As mentioned earlier, TWS from CLSM is obtained directly as one of the outputs, whereas TWS from NOAH is obtained as
the sum of the following components (Sun et al., 2019):

$$TWS = SnWE + SMC + CSWS \tag{1}$$

where SnWE represents snow depth water equivalent, SMC is soil moisture content, and CSWS is canopy and surface water storage. For both the aforementioned GLDAS products, the anomalies are computed by subtracting the long-term mean
monthly TWS of Jan 2004 to Dec 2009 from the monthly TWS values. Let the predictand variable GRACE TWSA be denoted
by $t$ and the set of predictors be denoted by $X$, the regression problem may be expressed as

$$t = f(X, p) \tag{2}$$





where $f$ represents the regression model and $p$ denotes the model parameter to be solved using $\{X_i, t_i\}_{i=1}^{N}$ as training data, where $i = 1...N$ is the index of training sample. $X$ includes CTWSA, NTWSA, and meteorological variables (P and T) for the current month, one month prior, and two months prior, along with three climate indices (DMI, NAO, ONI) for the current month. In this context, P, P_1, and P_2 represent precipitation for the current month, one month prior, and two months prior, respectively. Four ML algorithms (CNN, SVR, ETR and SER) are trained to solve the regression problem described in Eqn. 2. ML models built after training and validation can be used to simulate GRACE like TWSA using the inputs only. Fig.1 illustrates the overall workflow adopted in this study. An overview of the feature selection technique and a brief description of the ML models used in this study are presented in the subsections below.

### 3.1 Feature Selection using Bayesian networks (BN)

Among the multiple features (predictors) mentioned in the previous section, the most relevant features for simulating TWSA are selected utilizing the potential of Bayesian Networks (BN). BNs serve as compact representations of probabilistic relationships among a defined collection of random variables (Das and Chanda, 2022). These networks are characterized by a graph $G = (V, E)$, where each vertex (nodes) v $\epsilon$ V corresponds to one of the aforementioned random variables in X. Through edges (arcs) e $\epsilon$ E, the network articulates the conditional independencies or dependencies that exist among the variables within X, collectively termed the graph's dependence structure. This framework employs directed acyclic graphs (DAGs) to concisely encapsulate probabilistic relations and effectively portray the joint distribution of the variables (Scutari and Nagarajan, 2011). In the context of a DAG, the inclusion of arcs may or may not indicate a causal relationship between variables where one variable can be understood as the cause, and the other as the effect (Sevinc et al., 2020). When an edge connects two nodes in the graph, the node from which the edge originates is referred to as the parent node, while the node where the edge terminates is known as the child node. The process of determining the topology of the graph $G$ is termed structure learning. This involves identifying the graph structure that effectively represents the conditional independencies observed within the data. Several algorithms have been presented in the literature to tackle this problem, which is broadly classified into three categories: constraint-based (which are based on conditional independence tests), score-based (which are based on goodness-of-fit scores) and hybrid (which combines the previous two approaches). In the realm of continuous variables, score-based algorithms have outperformed their constraint-based counterparts in terms of performance. The challenge with constraint-based algorithms lies in their tendency to yield partially directed acyclic graphs (PDAG), which can subsequently disrupt precise predictions for the target variable (Das and Chanda, 2020).

Score-based learning offers both precise and approximate solutions. Precise solutions ensure the identification of the graph that optimizes an objective function while adhering to a maximum in-degree constraint (Constantinou et al., 2021). These algorithms, also referred to as search-and-score algorithms, involve the utilization of heuristic optimization methods to tackle the task of acquiring the structure of a Bayesian network. In this process, each potential network configuration is assigned a score that indicates its goodness of fit and the objective of the algorithm is to maximize this score. Hill-Climb (HC), a classical heuristic search algorithm has been the major choice for this purpose in most of the hydrology related studies (Vitolo et al., 2018; Das and Chanda, 2022, 2020; Dutta and Maity, 2021; Chanda and Das, 2022).

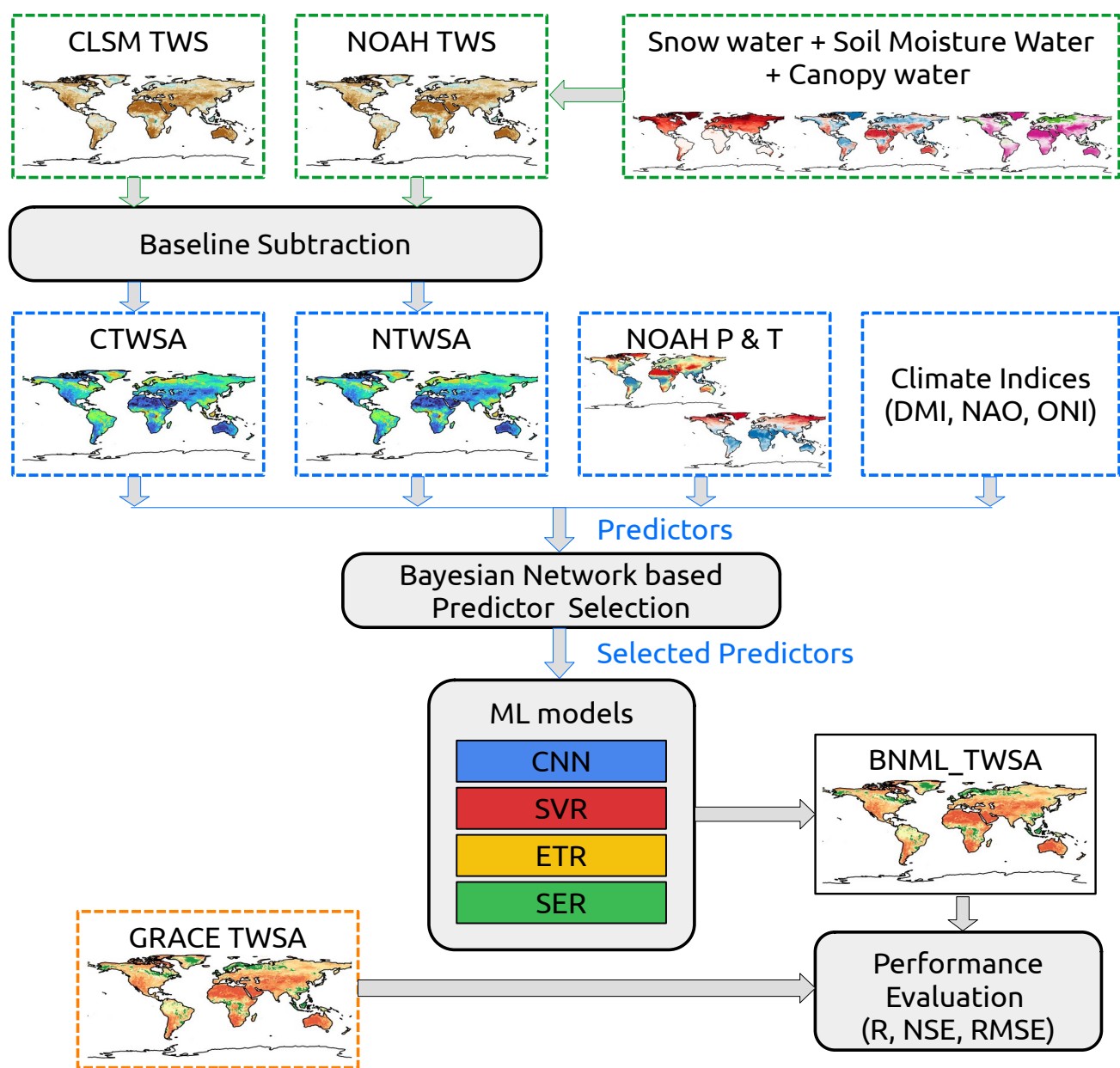

**Figure 1.** An overview of the methodology employed in this study





The Hill-Climbing (HC) algorithm commences its process with an initial empty graph and systematically explores potential DAGs to maximize the network's score, achieved through operations like adding, removing, or reversing arcs, determined by the strength of edges (Scutari and Denis, 2021). The network's score is calculated using the Bayesian Information Criteria (BIC) to prevent overfitting, wherein the BIC score is determined by a formula incorporating the number of parameters in the global distribution ('d') and the dataset length ('N'). A critical concept within the HC algorithm is edge strength, reflecting the disparity in the BIC score when a specific arc is included or excluded, indicating the degree of interdependency between connected nodes. A higher edge strength signifies a stronger correlation. In essence, the Hill-Climbing algorithm employs an iterative approach, optimizing the network score by manipulating the DAG's structure based on edge strengths determined through the BIC score, thereby striking a balance between model complexity and data fidelity while emphasizing significant node interconnections. The BIC score formula is given by:

$$BIC = \sum_{i=1}^{N} \log \left( P(X_i \,|\, MB(X_i)) \right) - \frac{d}{2} \log(N) \tag{3}$$

where MB represents the Markov blanket which is the minimal set of nodes that can predict a target node. The bnlearn package in the R environment (Scutari, 2010) is used to develop the DAG networks.

### 3.2 Machine Learning Algorithms for TWSA Modeling

This section presents a concise overview of machine learning algorithms used to model TWSA. In this study, three types of machine learning algorithms have been used: neural network-based (CNN), kernel-based (SVR), tree-based (ETR), and stacking ensemble regression (SER).

#### 3.2.1 Convolutional Neural Network (CNN)

CNNs belong to a category of neural networks particularly well-suited for handling data with grid-like structures, such as images or time-series data. The primary advantage they offer over conventional feed-forward neural networks is their utilization of mathematical linear operations known as convolutions (Uz et al., 2022). These layers possess the capability to autonomously extract features, identifying crucial aspects within the input data essential for establishing the correlation between input and output variables. Hence, CNNs have the capacity to manage raw data devoid of the necessity for preprocessing or manual extraction of features (Ferreira and da Cunha, 2020). CNNs find widespread application in image recognition tasks. For images, which possess two dimensions, convolutional filters of corresponding dimensions are employed. Conversely, for tasks involving sequential data or time series, like the context of this study, CNNs with one-dimensional (1D) convolutional filters (1D-CNNs) are employed. In this study, a 1D-CNN comprises a single convolutional layer and two fully connected layers. The activation function 'ReLU' is applied within the convolutional layer (Ferreira and da Cunha, 2020; Alibabaei et al., 2021; Ahmed et al., 2022). To mitigate overfitting, a dropout layer is introduced following each convolutional layer. The training algorithm employed for this model is Adam. The learning rate is established at 0.1, and the number of training epochs is determined using early stopping, with a maximum limit of 200 epochs. Additionally, the batch size is configured to 32.





### 3.2.2 Support Vector Regression (SVR)

Support Vector Regression (SVR) is a pivotal component of Support Vector Machines (SVM), an algorithm introduced by Cortes and Vapnik (1995), designed to handle non-linear regression problems. SVR extends the fundamental concept of SVM by effectively addressing regression tasks through a non-linear mapping of input data into a higher-dimensional feature space. The underlying principle of SVR is based on the concept of the "kernel trick". This technique employs a kernel function, such as the Radial Basis Function (RBF) kernel, to transform the input data into a feature space where linear regression can be applied effectively. The kernel function aids in defining a hyperplane—a decision boundary—within this feature space. This hyperplane facilitates the prediction of target values by distinguishing between different types of data patterns. Moreover, SVR aims to establish a boundary layer at a certain distance from the hyperplane, enclosing the data points that lie proximate to the hyperplane, known as support vectors. Selecting an appropriate kernel function is a crucial step in SVR. Raghavendra and Deka (2014) highlight that for non-linear problems, polynomial and RBF kernels are commonly employed. However, Das and Chanda (2020) advocate for the superiority of the RBF kernel in non-linear regression tasks, leading to its preference. For a more comprehensive understanding of the algorithms employed in this context, detailed insights and further elucidation can be gleaned from the work of Raghavendra and Deka (2014); Das and Chanda (2020); Das et al. (2022).

### 3.2.3 Extra Trees Regressor (ETR)

In the realm of ensemble-based predictive modeling, the ETR shares a fundamental principle which is build on the foundation of decision trees and random forests, combining their strengths to create an ensemble of diverse decision trees and is less likely to overfit a dataset (Ahmad et al., 2018). In the ETR framework, a random subset of features is employed to train each individual base estimator, akin to the approach used in Random Forest, where all the predictors are employed. This attribute ensures diversity among the constituent trees, contributing to the model's generalization capabilities and mitigating the risk of overfitting. Whereas Random Forest selects the optimal feature from the random subset to split a node, ETR takes a step further by introducing an additional layer of randomness (Kumar et al., 2022). ETR not only randomly selects a feature from the subset but also stochastically chooses the corresponding split value for the chosen feature. This distinctive feature selection strategy imparts an extra level of variance to the model, rendering it more robust and capable of capturing intricate relationships in the data (Ahmad et al., 2018). The predictions of the individual trees are combined to generate the ultimate prediction through a process of arithmetic averaging. The algorithm is influenced by two pivotal parameters: the count of predictors chosen randomly at each node and the minimum sample size required to initiate a node split (Sun et al., 2021).

### 3.2.4 Stacking Ensemble Regression (SER)

Stacking, a form of meta-learning, aims to enhance predictive performance by combining predictions from multiple base models through a higher-level integrated model (Zounemat-Kermani et al., 2021). The stacking generalization framework presents a couple of avenues for maximizing predictive gains. One approach involves employing diverse base learners, which fosters variability among the base models. Alternatively, enhancing ensemble size while keeping the number of base learners





constant can also provide the meta-learner with a broader range of insights. In this context, the term 'meta-learner' refers to
an aggregating model that learns the optimal way to combine outputs from the base learners. These 'base learners' constitute
the models whose individual predictions are assembled in the final step (Lee and Ahn, 2021). To mitigate overfitting, out-of-
sample data is employed for training the meta-model. This entails utilizing predictions from the base learners on this external
data. The overarching objective is for the meta-model to establish an optimal correlation between observed values and its own
predictions. The process is aptly termed 'stacking' since it involves merging predictions from validation sets, thereby creating
a fresh dataset for the meta-model to glean insights from. Additional valuable insights can be obtained from the comprehensive
analysis of ensemble machine learning presented in the review of Martinez-Gil (2022); Zounemat-Kermani et al. (2021); Lee
and Ahn (2021). In the present study, CNN, SVR and ETR are used as the base models. During the training process, initial
predictions are first generated by the base learners. Subsequently, these predictions from the base learners serve as inputs to the
meta learner, which produces the ultimate output. While all base models are utilized as potential meta models to assess overall
accuracy, preference is given to the Genralized Linear Model (GLM) model due to its superior effectiveness as a meta learner
compared to the alternative models.

### 3.3    Training and Performance Evaluation

The period of observed GRACE data used in this study is April 2002 to December 2022 (216 months in total excluding the
GRACE data gaps). Within this period, the available months from 2010 to 2016 (68 months) are used as the testing period
and the remaining part of the dataset (148 months) is used to train the ML models. The five-fold cross-validation technique
is employed during the training phase to address the issue of insufficient data length. Additionally, all input parameters for
the machine learning models have been normalized within the range of 0 to 1. The simulated TWSA employing BNs as the
optimal feature selector in conjunction with various ML models is henceforth referred to as BNML_TWSA. The performance
of the BNML_TWSA from each of the models is evaluated against GRACE/GRACE-FO TWSA through several agreement
metrics, including the Pearson correlation coefficient (CC), Nash-Sutcliffe efficiency coefficient (NSE), and root mean square
error (RMSE):

$$CC = \frac{\sum_{i=1}^{n}(O_i - \bar{O})(P_i - \bar{P})}{\sqrt{\sum_{i=1}^{n}(O_i - \bar{O})^2}\sqrt{\sum_{i=1}^{n}(P_i - \bar{P})^2}}, CC \in [-1, 1] \tag{4}$$

$$NSE = 1 - \frac{\sum_{i=1}^{n}(O_i - P_i)^2}{\sum_{i=1}^{n}(P_i - \bar{P})^2}, NSE \in (-\infty, 1] \tag{5}$$

$$RMSE = \sqrt{\frac{1}{n}\sum_{i=1}^{n}(O_i - P_i)^2}, RMSE \in [0, +\infty) \tag{6}$$





where, $O_i$ and $P_i$ represents the TWSA from GRACE/ GRACE-FO and the BNML_TWSA respectively, with $\bar{O}$ and $\bar{P}$ respec-
tively denoting their mean. The size of the test sample is denoted by $n$. A higher value for CC and NSE, closer to 1, as well as
a lower value for RMSE, closer to 0 indicates superior performance (Nash and Sutcliffe, 1970).

## 4   Results and Discussions

### 4.1   Selected predictors using BN

The optimal predictors for each grid across the globe (58027 grids in total) are selected using BN. For each predictor, the spatial
map showing the grids where they are selected by the BNs as optimal predictors is depicted in Fig. 2. Fig. 2 also shows a bar
plot with the number of grids where each predictor is retained by the BN. CTWSA is selected by BN as an optimal predictor
for maximum number of grids (71.06%), followed by ONI (56.03%) and NTWSA (48.71%). The selection of CTWSA as the
best predictor of GRACE TWSA is also suggested by other studies conducted in different regions around the globe Sun et al.
(2021). Among the 11 global river basins investigated specifically in this study, the number of grids with CTWSA as an optimal
predictor is lower in the Amazon and Nile basins compared to the other basins. Only 3.55% of the total grids holds the NAO
as an optimal predictor as selected by the BNs. This is reflected in the river basin scale as well; for each of the river basins, a
very limited percentage of grids retain the NAO as an optimal predictor.

**Figure 2.** Spatial distribution and bar plot of selected predictors using Bayesian network





## 4.2 Grid specific leader models

The predictors selected by the BNs in each grid are used as input to predict the TWSA using the four ML algorithms mentioned earlier: CNN, SVR, ETR, and SER. The grid wise leader ML algorithm is identified based on the Pearson correlation coefficient (R) between predicted TWSA and GRACE TWSA for the test period. Fig. 3 depicts the spatial distribution of the leader algorithms over the globe along with frequency as bar plot. ETR performs the best for the maximum number of grids, with a total of 25703, followed by SVR, SER, and CNN, which perform best for 11609, 11069, and 9646 grids respectively. Thus, for most of the river basins including Krishna and Godavari in India, Danube in Europe, Nile, Zambezi and Limpopo in the African continent, Mississippi in the USA and the transboundary GBM and Indus, ETR emerges as the leader model in maximum grids. The contribution of the leader algorithm as a percentage of the total grid points for each river basin is shown in Fig. 3c. It is observed that in the Limpopo river basin, ETR performs best in 89.0% of the grid points, whereas CNN does not perform best in any of the grids in this basin. In the Murray-Darling river basin in Australia, the four ML algorithms show the best performance at approximately equal number of grid points (CNN: 25.9%, SVR: 21.4%, ETR: 26.1% and SER: 26.6%).

## 4.3 Performance evaluation of simulated global BNML_TWSA

For the leader ML models at each grid, the BNML_TWSA is evaluated against the GRACE TWSA during the testing period (68 months). Performance measures such as the CC, NSE, and RMSE are computed for the BNML_TWSA at both the basin-wide and grid-wise levels. Similar performance measures are also computed for CTWSA and NTWSA. The spatial distribution of CC and NSE obtained from NTWSA, CTWSA and the BNML_TWSA using the identified leader ML models during the test period is shown in Fig. 4. According to Fig. 4a and 4c, it is evident that the agreement between CTWSA and GRACE TWSA is better than that of NTWSA. However, the BNML_TWSA (as shown in Fig. 4e) performs better in most grids worldwide compared to the TWSA obtained from the LSMs (CTWSA and NTWSA). The BNML_TWSA showed clear improvement in performance over NTWSA and even CTWSA in all the basins, except for the western part of the Nile basin and the southwestern part of the Mississippi basin where CTWSA shows a closer match with the GRACE TWSA in some grids. As shown by the BNML_TWSA results, the arid and semi-arid regions (for example, the arid desert part of Nile and the semi-arid regions of Mississippi) have poorer model performance, which is consistent with other global studies Mo et al. (2022). A substantial improvement in the performance by the proposed model can be observed in most parts of India, Eastern Europe, and South America. Fig. 4b, 4d and 4f depicts the NSE values obtained from NTWSA, CTWSA and the BNML_TWSA respectively. The superior performance of the proposed model on a global scale is evident from these plots.

The gridwise CC, NSE and RMSE values are further compared for the three TWSA datasets (NTWSA, CTWSA, and BNML_TWSA) in all river basins as boxplots depicted in Fig. 5. The median values of each matrix are listed in Table 2. For most of the basins, BNML_TWSA shows a higher median value of CC and a smaller range in the boxplot compared to NTWSA and CTWSA, indicating a better performance for the proposed BNML_TWSA. However, in Indus, Limpopo, Nile, and Zambezi, CTWSA has a slightly better performance than BNML_TWSA. All the grids in the Amazon, Danube, Godavari, Krishna, Mississippi and Zambezi basins have a CC value of $\geq 0.75$ for BNML_TWSA. Similarly, for GBM and Murray-

**Figure 3.** a) Frequency, b) spatial distribution of leader machine learning algorithms and c) leader machine learning algorithms in terms of percentage for different river basins



Darling, the lowest CC value among all grids is above 0.5. NSE, which compares the residual variance (the "noise") with the variance of the measured data, shows poor results ($< 0$) for most of the basin by both NTWSA and CTWSA. On the other hand, the NSE values for BNML_TWSA are fairly high ($> 0.68$) in most basins except for Indus, Limpopo, Murray-Darling and Nile, and only Indus has a median NSE value less than 0. Hence, based on the NSE value, BNML_TWSA has a higher

325 performance than NTWSA and CTWSA for all the river basins. The improved performance of BNML_TWSA can also be illustrated by the distribution of RMSE values over each river basin depicted in Fig. 5. When comparing the overall spread of RMSE values by BNML_TWSA, CTWSA, and NTWSA, it is observed that the range of RMSE values for BNML_TWSA is lower than that of CTWSA and NTWSA. This indicates that the interquartile range (IQR), the difference between the third quartile and the first quartile, of RMSE values for BNML_TWSA is smaller than that of CTWSA and NTWSA.

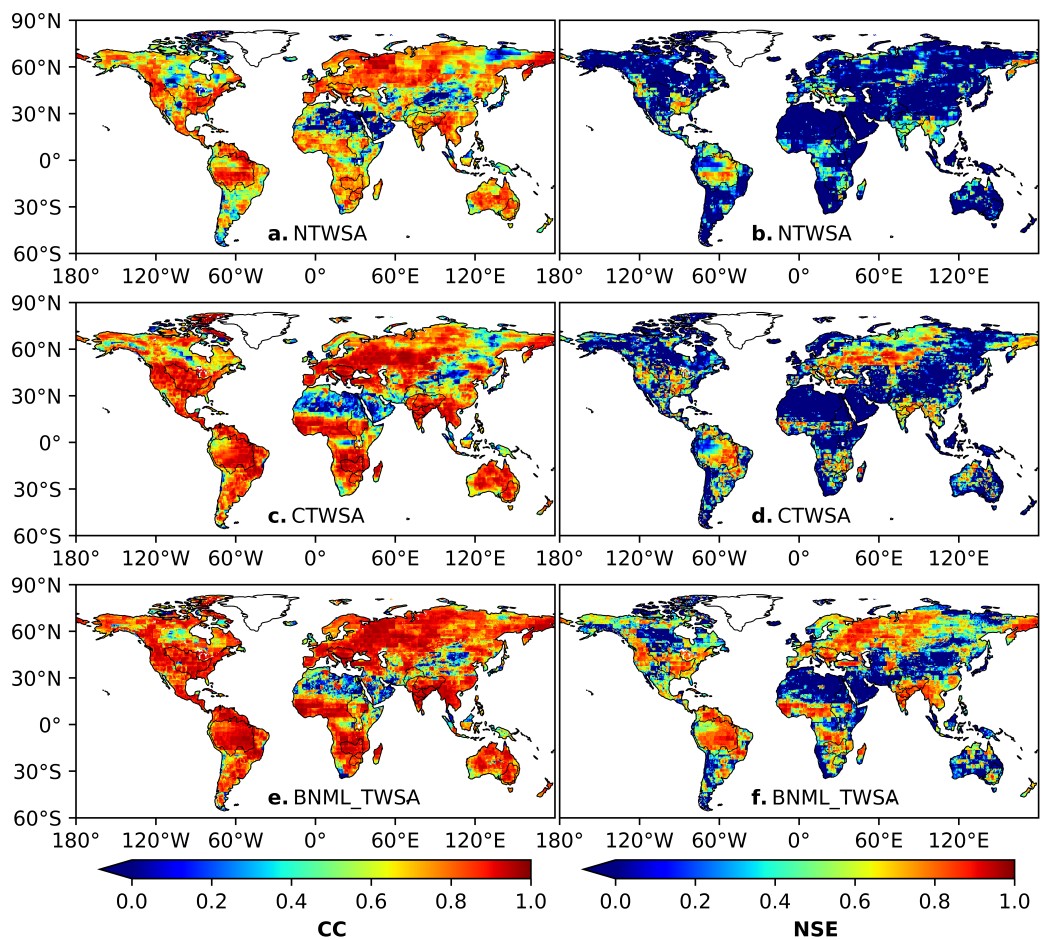

**Figure 4.** Correlation coefficients and NSE, between observed GRACE TWSA and NTWSA, CTWSA and BNML_TWSA.



**Figure 5.** Box plot of CC, NSE and RMSE values for the grids of each basin, excluding the outliers.





**Table 2.** Median of CC, NSE and RMSE values at grids of each basin. A bold value signifies the best performance.

| Basin | CC | | | NSE | | | RMSE | | |
|---|---|---|---|---|---|---|---|---|---|
| | NTWSA | CTWSA | BNML_TWSA | NTWSA | CTWSA | BNML_TWSA | NTWSA | CTWSA | BNML_TWSA |
| Amazon | 0.822 | 0.876 | **0.910** | 0.537 | 0.608 | **0.796** | 119.5 | 107.8 | **83.3** |
| Danube | 0.806 | 0.917 | **0.918** | 0.433 | 0.700 | **0.766** | 56.0 | 42.0 | **35.9** |
| GBM | 0.766 | 0.861 | **0.879** | 0.266 | 0.452 | **0.699** | 123.0 | 100.7 | **74.5** |
| Godavari | 0.779 | 0.898 | **0.927** | 0.475 | 0.671 | **0.839** | 114.4 | 86.0 | **63.7** |
| Indus | 0.566 | **0.654** | 0.651 | -1.140 | -0.392 | **-0.003** | 51.7 | 46.4 | **37.1** |
| Krishna | 0.740 | 0.895 | **0.924** | 0.424 | 0.636 | **0.810** | 102.3 | 81.3 | **63.0** |
| Limpopo | 0.800 | **0.839** | 0.837 | -0.897 | 0.351 | **0.581** | 54.4 | 34.0 | **30.5** |
| Mississippi | 0.786 | 0.903 | **0.907** | 0.105 | 0.539 | **0.704** | 61.4 | 50.1 | **39.9** |
| Murray-Darling | 0.788 | 0.853 | **0.865** | -0.897 | -0.060 | **0.455** | 57.3 | 43.6 | **35.0** |
| Nile | 0.538 | **0.739** | 0.675 | -1.373 | -1.570 | **0.200** | 61.6 | 55.1 | **31.7** |
| Zambezi | 0.770 | **0.925** | 0.923 | 0.159 | 0.651 | **0.688** | 123.3 | 75.1 | **70.0** |

## 4.4 Basin scale quality assessment of gap-filled TWSA

The results presented in the preceding sections demonstrate the superior ability of the proposed model to simulate GRACE TWSA during the testing period. The leader model, constructed for each global grid, is utilized to generate a GRACE-like TWSA series from April 2002 to December 2022 using the grid-wise input parameter set selected by the BNs. Fig. 6 shows the time series of average reconstructed TWSA for all grids within a river basin. TWSA from GLDAS NOAH Rodell et al. (2004) (NTWSA), GLDAS CLSM Li et al. (2019) (CTWSA), and GRACE are also included in Fig. 6 for comparison. The seasonal variation in TWSA is greatly captured by the proposed models and the other two LSM outputs. Similar to observations from previous sections, the CTWSA has better performance than the NTWSA. However, even better performance is achieved by the proposed BNML_TWSA. Compared to the observed GRACE TWSA, both NTWSA and CTWSA underestimate the peak values of TWSA for the Amazon river basin, whereas BNML_TWSA time series matches the GRACE TWSA most closely. For the Indus and Nile river basins, the mismatch between GRACE TWSA and TWSA from LSM outputs has been widening since 2010; however, the proposed model simulates BNML_TWSA, which is very close to the GRACE TWSA.

The basin-wise performance of mean BNML_TWSA and CTWSA is assessed against the GRACE TWSA and depicted as a scatter plot in Fig. 7. NTWSA is not included in this plot as it is already found to be a poorer match with GRACE TWSA than the CTWSA. The CC of BNML_TWSA in Zambezi is the highest among all the basins (0.989), however the RMSE (46.1 mm) and NSE (0.833) value does not suggest its performance as the best. Basin-wide mean BNML_TWSA has higher CC ($\geq 0.904$) than CTWSA for GBM, Godavari, Indus, Krishna, Limpopo, Mississippi, Murray-Darling and Zambezi basins (see Fig. 7). In the Nile basin, a high CC value of 0.82 is obtained for BNML_TWSA, however, this is the minimum among all river basins. The Nile basin is one of the few basins where CTWSA has a slightly higher CC (0.889) than BNML_TWSA (see Fig. 7), but





the time series plot (see Fig. 6) shows that the BNML_TWSA is in better agreement with GRACE TWSA. This indicates that,

in general, the proposed BNML_TWSA is more reliable than CTWSA. Considering NSE and RMSE values, Amazon, GBM, Godavari, Krishna and Nile basins perform better with BNML_TWSA than with CTWSA. The only exception is Danube basin which has a marginally lower performance for BNML_TWSA than CTWSA according to basin-wide mean.

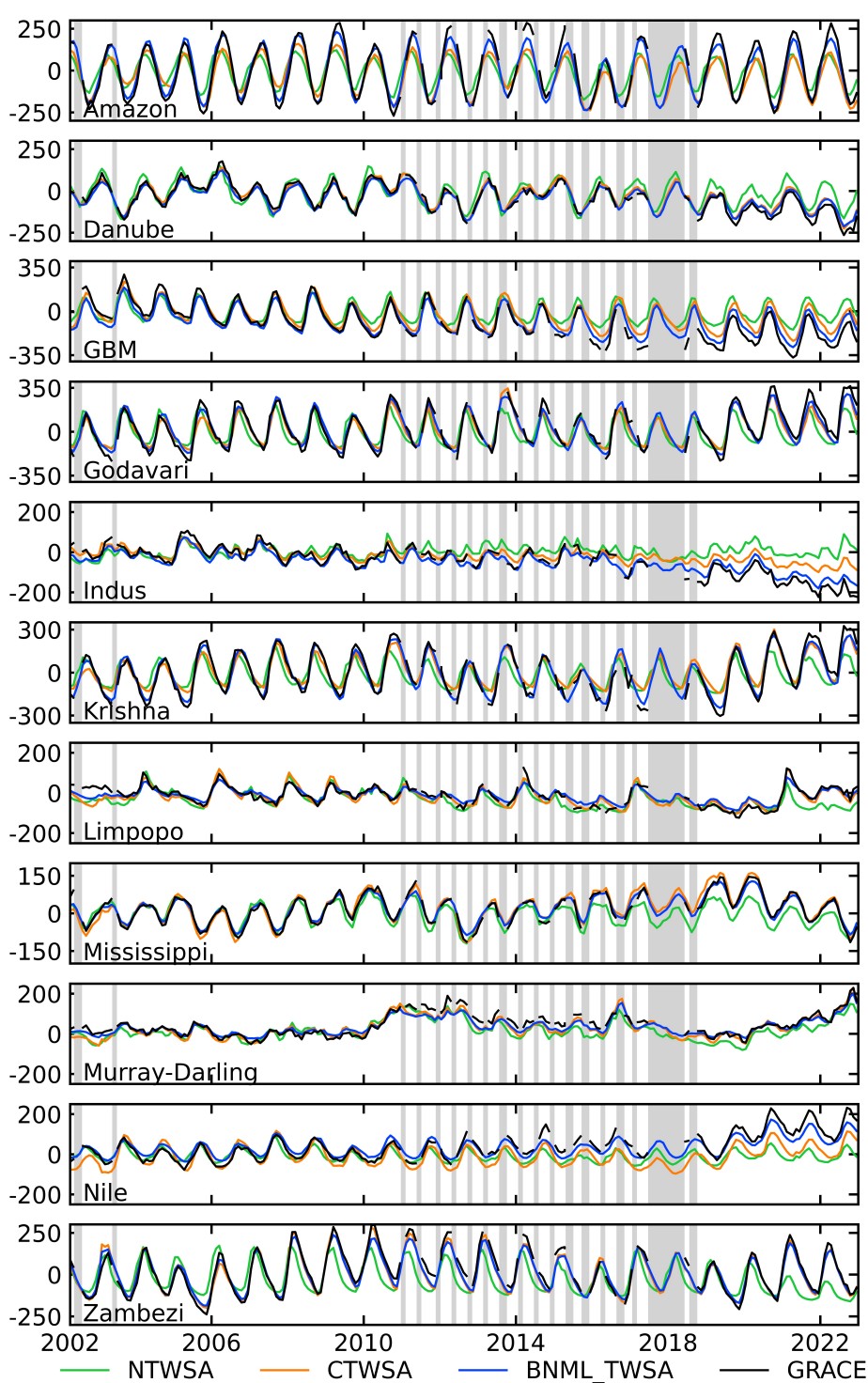

**Figure 6.** Comparison of TWSA time series from April 2002 to December 2022 (GRACE period)



**Figure 7.** Scatter plot of basin-wise mean of observed GRACE TWSA vs BNML_TWSA and CTWSA of each river basin





### 4.5 Reflection of hydroclimatic extreme events in BNML_TWSA during pre-GRACE period (1960-2002) and intermediate gap periods

The leader ML model of each grid is used to hindcast BNML_TWSA during the pre-GRACE period (Jan 1960 - Mar 2002) using the predictors selected by BNs. Basin averaged BNML_TWSA series is shown in Fig. 8. Similar to the LSM outputs (CTWSA and NTWSA), increasing and decreasing trends are noted for BNML_TWSA series for some of the river basins. The variability of TWSA hindcast is lowest for Indus (-95.1 to 75.4 mm) and Limpopo (-61.6 to 83.7 mm) river basins, while Amazon (-227.5 to 199.6 mm) and Krishna (-200.7 to 239.5 mm) have the highest variability. The minimum TWSA hindcast
values for Murray-Darling and Nile are around -20 mm, which is the lowest among all the regions. During this hindcast period, a number of significant climate extreme events have occurred; for example, two severe flood events of India are depicted in Fig. 9a-b. The Gomti River, a tributary of the Ganges River, overflowed and caused a severe flood that inundated half of the city of Lucknow in mid October 1960. As shown in Fig. 9a, water storage increased around Lucknow in October 1960, which is marked by a green rectangle. On 11 August 1979, a flood disaster occurred in Gujarat, India, when the Machchhu dam failed
and submerged the town of Morbi, killing about 1500 people Saharia et al. (2021). The deviation of BNML_TWSA from the long-term monthly mean for the Morbi region in Gujarat, which is marked by a green rectangle, is shown in Fig. 9b. Results clearly indicates the enhancement of terrestrial water storage during this period. The proposed model accurately reproduced the TWSA series during the pre-GRACE period and detected recorded climate extreme events that occurred in the hindcast period.

During the GRACE gap period, several climate extreme events occurred around the world. In the CONUS region Hurricane Harvey is one of these climate extreme events which made landfall on 25th August 2017 along Texas and Louisiana Gulf Coast area. This catastrophic flood event caused more than 100 deaths and damage of 125 billion US dollars Sun et al. (2021). The flood event is expected to reveal an increase in TWSA compared to the long-term mean TWSA of that region. This is well reflected in Fig. 9c which depicts the difference between BNML_TWSA for September 2017 (as the event occurred towards
end Aug 2017) and the long-term mean TWSA for September. Similarly, heavy rain on July 15-16, 2017 led to flooding in several districts of Gujarat, India and the event reportedly caused more than 200 deaths. This is depicted with a similar plot in Fig. 9d. The proposed model effectively reflected the impact of climate extreme events on terrestrial water storage in the Texas and Louisiana Gulf Coast area and Gujarat in the map.

**Figure 8.** BNML_TWSA during the pre-GRACE period (Jan 1960 - Mar 2002)

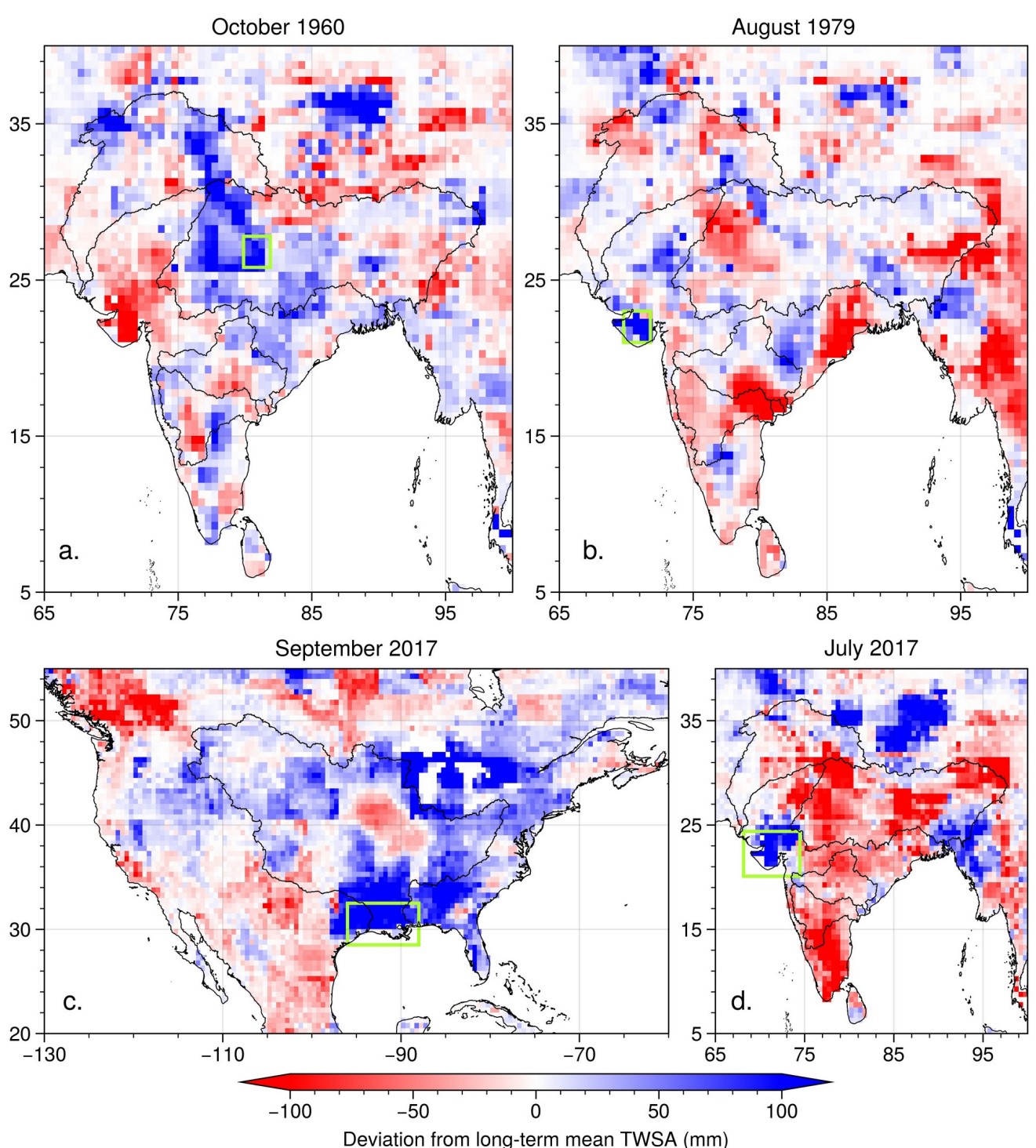

**Figure 9.** Difference between monthly and long-term mean monthly BNML_TWSA reflecting hydroclimatic extreme events. The zone of interest is marked by a green rectangle.




## 4.6 Comparison with previous studies

In this section, the reconstructed BNML_TWSA is compared with the global TWSA products developed recently by Humphrey and Gudmundsson (2019) and Sun et al. (2020). The reconstructed TWSA datasets developed using statistical models by Humphrey and Gudmundsson (2019) are named GRACE-REC. Two GRACE-REC datasets, which include monthly ensemble mean data from JPL-MSWEP and JPL-ERA5 datasets respectively, and the BNML_TWSA, are each evaluated against the GRACE JPL mascon dataset. The grid intersection points and resolution (0.50°×0.50°) of BNML_TWSA, GRACE-REC, and

GRACE JPL Mascon are uniform, which eliminates the requirement for regridding. The period of comparison is selected as the common available dataset duration of the above three products. Fig. 10 depicts the spatial distribution of the CC values obtained from two GRACE-REC dataset of Humphrey and Gudmundsson (2019) along with corresponding CC values of the BNML_TWSA each compared with GRACE JPL mascon datasets. Fig. 11 shows NSE values similar to the Fig. 10. Based on Fig.11 and Fig. 10, it is evident that the performance of BNML_TWSA surpasses that of GRACE-REC TWSA. Notably, both

the GRACE-REC products exhibited sub-optimal performance in the region near the Sahara desert and Saudi Arabia.

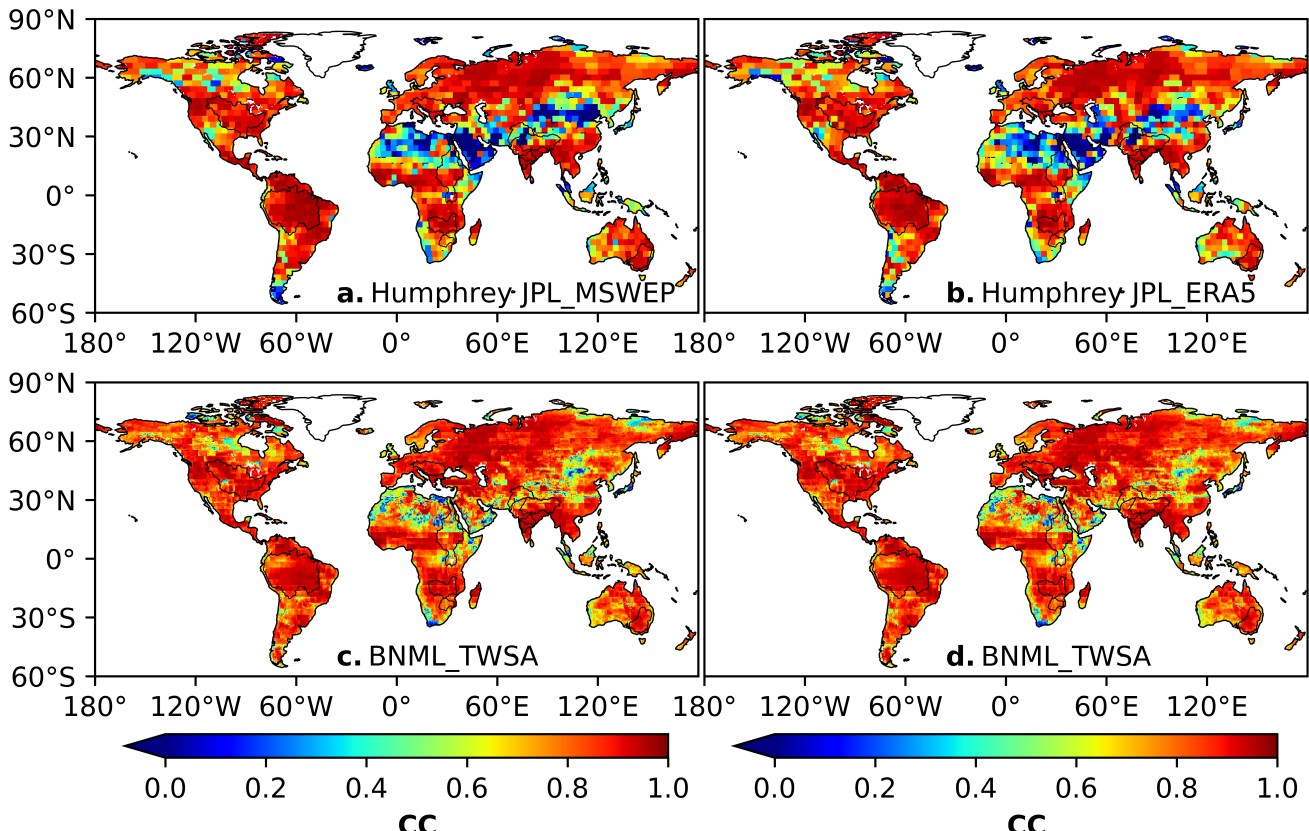

**Figure 10.** Comparison of correlation coefficient (CC) values obtained by two GRACE-REC products from Humphrey and Gudmundsson (2019) and BNML_TWSA, each evaluated against GRACE JPL mascon.

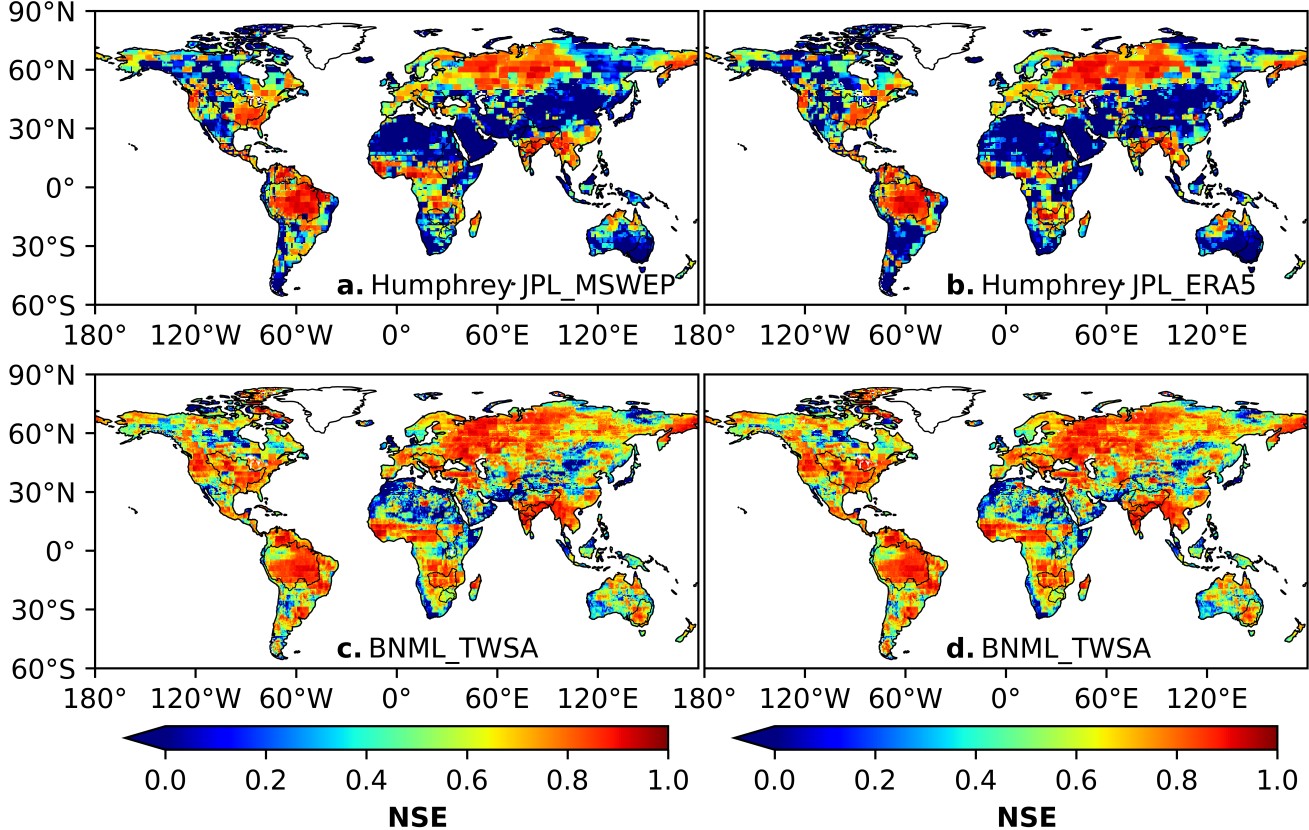

**Figure 11.** Comparison of Nash-Sutcliffe Efficiency (NSE) values obtained by two GRACE-REC products from Humphrey and Gudmundsson (2019) and BNML_TWSA, each evaluated against GRACE JPL mascon.

Next, we compare the agreement of BNML_TWSA with the observed GRACE JPL mascon versus the agreement of TWSA derived from Deep Neural Network (DNN) models (namely DNN_JPL-M and DNN_CSR-M) by Sun et al. (2020) with the same GRACE JPL mascon. During the development of the DNN_JPL-M and DNN_CSR-M TWSA products in Sun et al. (2020), the JPL mascon and CSR mascon GRACE products are respectively used as targets. These two particular products

of Sun et al. (2020) are selected for comparison as the study mentions that TWSA derived using DNN models demonstrated superior performance compared to the other two learning-based models attempted in their study. The spatial resolution of DNN_JPL-M and DNN_CSR-M is 1.0°×1.0°, whereas the spatial resolution of BNML_TWSA and JPL mascon is 0.50°×0.50°. To ensure uniform spatial resolution for all TWSA products, both BNML_TWSA and JPL mascon are regridded (upscaled) to 1.0°×1.0°, similar to DNN_JPL-M and DNN_CSR-M. Fig. 12a and 12b depicts the CC and NSE values

respectively for DNN_JPL-M and similar indices for DNN_CSR-M are shown in Fig. 12c and 12d. CC and NSE values for BNML_TWSA are depicted in Fig. 12e and 12f respectively. The prediction accuracy of BNML_TWSA is the best when compared DNN_JPL-M and DNN_CSR-M TWSA.

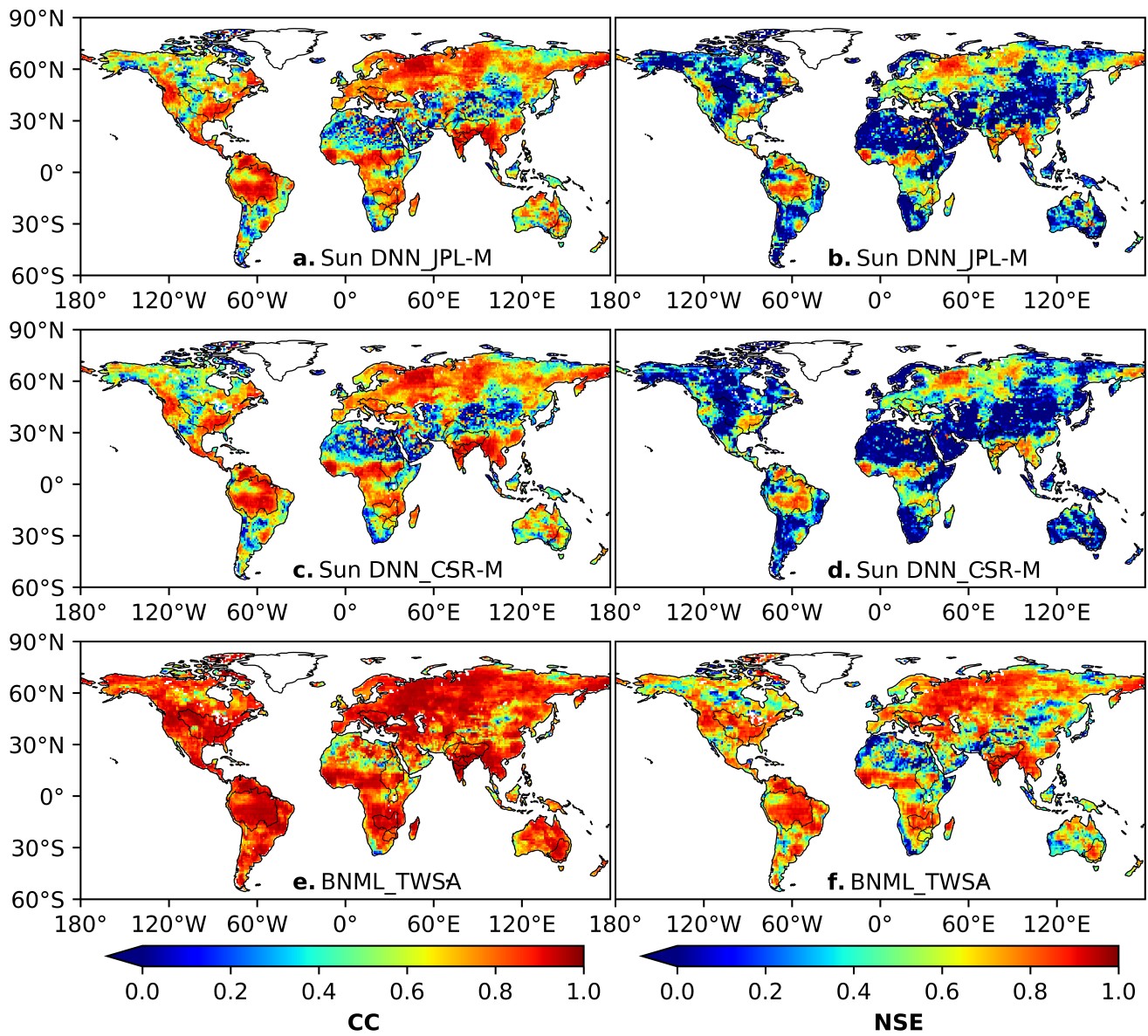

**Figure 12.** Comparison of correlation coefficient (CC) and Nash-Sutcliffe Efficiency (NSE) values obtained by two reconstructed TWSA products from Sun et al. (2020) and BNML_TWSA, each evaluated against GRACE JPL mascon.

The basin-wise median CC and NSE values of BNML_TWSA, the reconstructed TWSA from Humphrey and Gudmundsson (2019), and Sun et al. (2020) are compared in Fig.13 using a radar chart. Specifically, for Humphrey JPL_ERA5, the CC values are shown in Fig.13a, and the NSE values in Fig.13d. Similarly, for Humphrey JPL_MSWEP, the CC values appear in Fig.13b, and the NSE values in Fig.13e. The CC and NSE values of Sun DNN_JPL-M and Sun DNN_CSR-M, which




have a similar analysis period, are presented in Fig.13c and 13f, respectively. Thus, Fig.13a-c depicts the basin-wise median CC values for the mentioned models, while Fig.13d-f illustrates the basin-wise median NSE values after excluding those below zero. With the exception of the Danube, GBM, Indus, and Nile river basins, the median CC values for BNML_TWSA

and Humphrey JPL_ERA5 (as shown in Fig. 13a) exhibit similarity. BNML_TWSA demonstrates higher median CC values compared to Humphrey JPL_ERA5 across the aforementioned basins. Specifically, the improvements in CC values obtained with BNML_TWSA are as follows: from 0.86 to 0.90 for the Danube, from 0.80 to 0.90 for GBM, from 0.59 to 0.77 for the Indus, and from 0.70 to 0.77 for the Nile. Similarly, compared to Humphrey JPL_MSWEP (Fig.13b), BNML_TWSA provides improvements in CC values such as: from 0.75 to 0.9 at GBM and from 0.47 to 0.74 at the Indus. When compared

with the reconstructed TWSA products by Sun et al. (2020), BNML_TWSA surpasses their accuracy as the median CC-values are high across all the river basins (Fig.13c). In terms of median NSE values, BNML_TWSA outperforms Humphrey JPL_ERA5 in all the river basins. Specifically, for BNML_TWSA, the median NSE value improves from -1.07 to 0.89 at the Murray–Darling and from -0.17 to 0.77 at the Nile. Similarly, BNML_TWSA exhibits improved performance compared to Humphrey JPL_MSWEP across all the basins, particularly at Murray–Darling, where the median NSE improves from -0.24

to 0.72. While Sun DNN_JPL-M performs better than Sun DNN_CSR-M, BNML_TWSA consistently outperforms both the products of Sun et al. (2020) across all the river basins. Notably, at the Nile, the median NSE value improves from 0 to 0.53 with BNML_TWSA compared to Sun DNN_JPL-M.

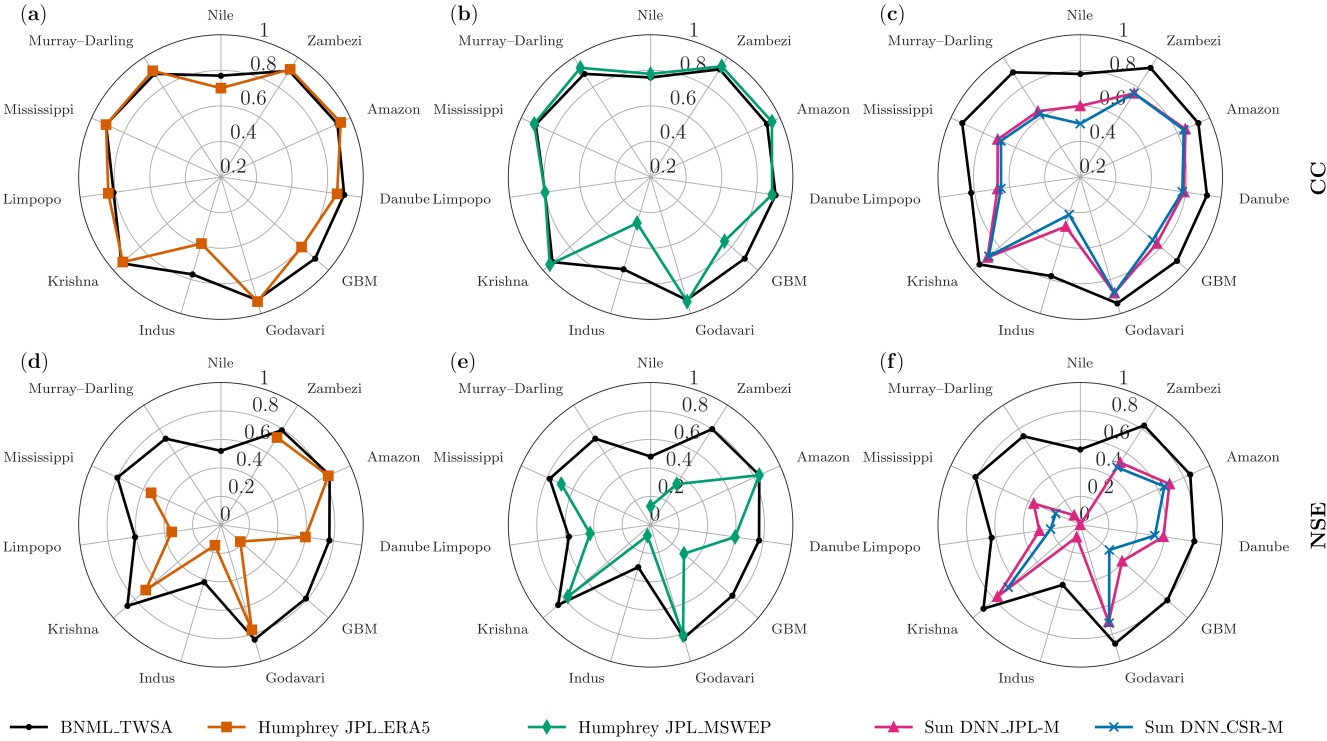

**Figure 13.** Comparison of median values of correlation coefficient (CC) and Nash-Sutcliffe Efficiency (NSE) across different basins. Note: Median NSE values below 0 are omitted during plotting.

## 5 Conclusions

In this study, we utilized Bayesian Networks (BNs), a novel feature selection technique, and machine learning (ML) models
to reconstruc global Terrestrial Water Storage Anomalies (BNML_TWSA), which is a GRACE-like TWSA dataset, thereby
filling data gaps in GRACE and generating hindcasts for the pre-GRACE period. The major conclusions from this study are
enumerated below.

For the target TWSA, optimal inputs are selected among meaningful predictor variables, such as land surface model outputs
(TWSA from Catchment Land Surface Model [CTWSA] and Noah Land Surface Model [NTWSA]), meteorological vari-
ables (Precipitation and Temperature), and climate indices (Dipole Mode Index [DMI], North Atlantic Oscillation [NAO], and
Oceanic Niño Index [ONI]). It is observed that the climate indices, ONI and DMI, are selected by BN as optimal predictors
for a large number of grids globally, along with TWSA from LSM outputs. This establishes that, in addition to the available
LSM based TWSA products, large scale climate indices are more important predictors of TWSA than the local meteorological
inputs.

At the global scale, Convolutional Neural Network (CNN), Support Vector Regression (SVR), Extra Trees Regressor (ETR),
and Stacking Ensemble Regression (SER) models are employed following the selection of optimal features through BNs at each





grid to finally obtain BNML_TWSA. It is noted that a single ML model cannot perform optimally across all grids worldwide, due to the significant spatial variability of important predictors. However, the performance of ETR is found to be the best for most of the grid points within the Ganga-Brahmaputra-Meghana, Godavari, Krishna, Limpopo and Nile river basin. ETR
performs best in 44% grids worldwide followed by SVR, SER and CNN.

The proposed approach yields a more reliable estimate of TWS compared to the outputs of global hydrological and land surface models (LSMs), which have significant biases due to inherent uncertainty and lack of representation of some of the physical processes. BNML_TWSA outperform the NTWSA and CTWSA for most grids worldwide. For the river basins such as Indus and Nile, BNML_TWSA matches GRACE TWSA very closely, even during the period when the TWSA from LSM
outputs deviates substantially from GRACE TWSA. While evaluating basin wise average BNML_TWSA against GRACE TWSA, Zambezi basin in Africa exhibited the highest correlation coefficient (CC=0.989), followed by Godavari (CC=0.984) and Krishna (CC= 0.983) in India. Further, the accurate reflection of historical climate extreme events, such as major floods, via the hincasted BNML_TWSA supports the enhanced accuracy of the proposed model and the developed TWSA dataset. A comparative analysis with TWSA products developed in recent literature (Humphrey and Gudmundsson, 2019; Sun et al.,
2020) indicates that BNML_TWSA surpasses these datasets when evaluated against the overlapping GRACE period. Hence, this study demonstrates that the proposed BN and ML based approach can effectively learn complex relationships between various inputs and GRACE TWSA, enabling global reconstruction and hindcasting of TWSA which is essential for several hydroclimatological studies.

*Data availability.* The presented dataset is published at https://doi.org/10.6084/m9.figshare.25376695 (Mandal et al., 2024) and updates
will be published when needed. All datasets utilised in this study are readily accessible; comprehensive dataset information, along with their respective links, is provided in this section. JPL GRACE Mascon data are obtained from Physical Oceanography Distributed Active Archive Center NASA/JPL (2023); Watkins et al. (2015) (https://podaac.jpl.nasa.gov/dataset/TELLUS_GRAC-GRFO_MASCON_CRI_GRID_RL06.1_V3). TWS data are retrieved from NASA GLDAS CLSM simulations Li et al. (2019) (https://doi.org/10.5067/LYHA9088MFWQ and https://doi.org/10.5067/TXBMLX370XX8). Canopy surface water, soil moisture content, snow water, precipitation, and temperature data
are taken from GLDAS Noah Land Surface Model Rodell et al. (2004) (https://doi.org/10.5067/9SQ1B3ZXP2C5 and https://doi.org/10.5067/SXAVCZFAQLNO). Dipole Mode Index is taken from the National Oceanic and Atmospheric Administration (NOAA) Physical Sciences Laboratory Saji et al. (1999); Saji and Yamagata (2003) (https://psl.noaa.gov/gcos_wgsp/Timeseries/Data/dmi.had.long.data). North Atlantic Oscillation index Wallace and Gutzler (1981); Barnston and Livezey (1987) (https://www.cpc.ncep.noaa.gov/products/precip/CWlink/pna/norm.nao.monthly.b5001.current.ascii) and Oceanic Niño Index Barnston et al. (1997) (https://www.cpc.ncep.noaa.gov/data/indices/oni.
ascii.txt) are retrieved from NOAA Climate Prediction Center.

*Competing interests.* The contact author has declared that none of the authors has any competing interests





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
