# Peer review of "Optimal feature selection for improved ML based reconstruction of Global Terrestrial Water Storage Anomalies"

_Earth System Science Data, 2024_

## Author Comment (AC1)

**Authors response to reviews for manuscript number #essd-2024-109**

The authors would like to thank the editor and reviewers for their constructive comments and suggestions that have helped improve the quality of this manuscript. The manuscript has undergone a thorough revision according to the reviewers' comments. Please see below our responses. For the reviewers' convenience, we have highlighted changes in the revised manuscript.

**Reviewer 1**

**RC 1.0:** The manuscript describes the derivation of a global long-term terrestrial water storage anomalies data-set, derived from a blending of GRACE satellite observations and global land surface models based on Bayesian networks and machine learning methods.
Such long term information about terrestrial water storage variations is valuable and can help to assess long term trends and to localize extremes. Therefore, I think the manuscript is relevant for publication, however in its current form it lacks to address uncertainties of the product and it is rather structured like a classic scientific study than a data description paper. The structure of the dataset itself is not suitable for efficient usage in the current form and needs revision. Thus for a publication in ESSD, my concerns are as follows.
**Response:** We thank the reviewer for considering this manuscript as relevant for publication. The concerns regarding the structure of the dataset and other observations are addressed in the following specific responses to the comments.

**RC 1.1:** From the title it does not become clear what dataset you would like to advertise. Should it be the TWSAs or the optimal features? I guess its the TWSAs that you would finally like to advertise so you should find a new title in the sense of "ML-based ... long-term terrestrial water storage anomalies from satellite and land-surface model data ...". The term "Optimal feature selection" does not generate an association with a data product, at least for me.
**Response:** We appreciate the suggestion of the reviewer for the title of the manuscript. We have incorporated a new title for our manuscript in the revised version as shown below:
"ML based reconstruction of long-term global terrestrial water storage anomalies from observed, satellite and land-surface model data"

**RC 1.2:** In the abstract, you write that you reconstruct TWSA but you don't specify the a grid type and the spatial resolution of the produced dataset. Do you only provide the gridded dataset or also basin aggregates? You should provide this information already in the abstract, although very briefly, so that the reader knows what to expect. Further, I suggest to add a section for data description that explains the structure and content of the final data product in the repository
**Response:** The abstract is updated with details of the spatial resolution and coverage of the developed dataset as below:
"The presented gridded dataset is published at `https://doi.org/10.6084/m9.figshare.25376695` (Mandal et al., 2024), featuring a spatial resolution of $0.50° \times 0.50°$ and offering global coverage."
During this manuscript review and discussion period, the global performance of the developed dataset is discussed in details in the manuscript; whereas, the actual dataset for all grid cells over the Mississippi river basin is made available in the repository for persual. After acceptance of the manuscript, the dataset for the entire globe will be made available in the same repository. Instead of adding a new section on 'data description', we have updated the 'Data availability' section with required details as below:
"The presented dataset is published at `https://doi.org/10.6084/m9.figshare.25376695` (Mandal et al., 2024) and updates will be published as and when needed. The BNML_TWSA dataset is available for all grid cells globally, with a spatial resolution of $0.50° \times 0.50°$, similar to the JPL GRACE Mascon, and is provided in NetCDF format."

**RC 1.3:** The term "optimal predictors" is mentioned in the title and introduction but the explanation in the methods section (3, 3.1) is not fully clear. What are the optimal predictors? Are they a subsection of your full predictors list?

Do you drop training data sets? Are the optimal predictors the ones that have the maximum impact (weight) in the ML algorithms? This should be made more clear and the benefit of knowing the optimal predictors should be outlined.

**Response:** The optimal predictors are a subset of all potential predictors of 15 variables. At each grid cell, out of the 15 variables, a subset is selected through BN, which serves as the "optimum set of predictors" for that grid cell. The training datasets are not dropped; rather, the "optimum set of predictors" are selected from the training period and are used subsequently for the next step of the analysis, which is the prediction of TWSA via ML algorithms. The "optimum set of predictors" are selected based on probabilistic independence/dependence structure; they do not indicate any impact/ weight on ML algorithms. This information is presented in section 3.1 of the revised manuscript.

In the introduction section benefit of knowing the optimal predictors is outlined as follows: "Selection of optimal predictors is not just a methodological novelty, but a critical step to ensure that the model prioritizes the most relevant and physically meaningful predictors. This approach reduces noise, minimizes overfitting, and enhances interpretability, making the final product more scientifically robust and practically useful (Das and Chanda, 2024)."

**RC 1.4:** You are not always consistent with your vocabulary, in 3.1 you introduce the term features for what you named previously predictors. I think you should keep a single notion here (and mention the term feature only once, maybe in brackets if this is needed because it's well known by the community).

**Response:** We thank the reviewer for highlighting the inconsistencies in the vocabulary used in our manuscript. In this revised version of the manuscript, we have addressed and rectified these inconsistencies, consistently using the term "predictors" throughout the manuscript.

**RC 1.5:** Reproducibility: for making the creation of your data set reproducible, you should at least mention which software tools you used for the machine learning and eventually publish the configurations alongside with your datatset or in another DOI based repository.

**Response:** We have used the standard python packages such as *tensorflow*, *keras*, *sklearn*, *xgboost* and *mlxtend* for building the machine learning models as well as *matplotlib* for generating the plots. No specific software tools for machine learning were used. In the final stage of publication, the python codes for building machine learning models will be made available in the same DOI based repository for reproducibility. This information is now provided in the revised manuscript in a newly added 'Code availability' section.

**RC 1.6:** The selection of evaluation metrics may not be ideal for the global evaluations. CC will always be high for regions with a clear annual amplitude whereas for the deserts with less variations and seasonality it is hard to get a good score in CC. NSE is especially designed for assessing peak flows. Maybe KGE would suit better here. And wouldn't a directed error metric like ME provide additional insight on over- or underestimation tendencies?

**Response:** Based on the reviewer's suggestion, we have included the Kling-Gupta Efficiency (KGE) metric alongside the previously used evaluation metrics.

**RC 1.7:** I think the introduction and the 4.6. Section could be shortened a bit in favor of a data(set) description section.

**Response:** As mentioned in an earlier comment, details about the presented dataset such as resolution and coverage are incorporated in the "Data availability" section in the revised manuscript. Moreover, for better usability, the dataset along with its metadata are now provided in the NetCDF format, which is the conventional format for all climate datasets including the JPL Mascon dataset. Regarding the introduction section,it has been shortened a bit as suggested. However, some more information about the motivation of the study is incorporated in that section to address the comment # RC 2.8 of reviewer #2.

**RC 1.8:** For many of the references DOIs are missing. For several DOI links are incorrect with duplicates in their URLs.

**Response:** The problem of missing DOIs and incorrect duplicate URLs in the references has been addressed, and all references have been thoroughly reviewed and updated with the correct DOIs where applicable.

**RC 1.9:** Your results should be evaluated in the light of uncertainties of GRACE based water storage anomalies, e.g.,

https://agupubs.onlinelibrary.wiley.com/doi/10.1029/2021JB022081; there are several different GRACE solutions available which have different levels of uncertainty (https://doi.org/10.1029/2023JB026908) so why did you select exactly one of them and how would the uncertainties of the GRACE product propagate into your BN_TWSA product? The characterization of uncertainties of your gridded and aggregated data sets would be important with respect to deriving any long term trends.

**Response:** In the revised version of the manuscript, we have incorporated a new section to discuss and assess the uncertainty involved in the BNML_TWSA, which is also presented below:

**Uncertainty, Limitations and future scope**

There are various sources that contribute to the uncertainties in reconstructed TWSA. The primary source of uncertainties arises from the inherent processing errors associated with the original GRACE data, as documented by Boergens et al. (2022) and Gao et al. (2023). Nevertheless, this issue is effectively mitigated by utilizing the mascon solution, which demonstrates clear superiority over the spherical harmonics data (Kalu et al., 2024). Another source of uncertainty stems from the machine learning models, which may be categorized into contributions from inadequacies and/or lack of knowledge regarding the model (epistemic) and data noise (aleatoric). In the present study, epistemic uncertainty has been reduced to some extent by training four different ML models at each grid cell and selecting the best-performing model to reconstruct the BNML_TWSA globally. On the other hand, Aleatoric uncertainty may arise from the input dataset i.e., the selected predictors. Analyzing the spatial distribution of selected predictors using BNs (Fig. 1), it becomes apparent that commonly employed forcing variables such as precipitation (P) and temperature (T) do not rank among the top predictors in most grid cells. This observation suggests that these forcing variables are already accounted for in the Land Surface Models (LSMs) as indicated by Sun et al. (2019). However, these variables are still selected as optimal predictors in some of the grid cells, which implies that physics-based LSMs may not entirely capture the total information encapsulated in the raw data. Consequently, incorporating a diverse set of variables—including those already utilized in physics-based LSMs—as potential predictors could mitigate model structural errors and parameter uncertainties inherent in the LSMs (Sun et al., 2020, 2019). Furthermore, uncertainties may also depend on the actual source of the input variables. For example, precipitation from satellite sources will entail different uncertainties compared to LSM based precipitation. In the present study, aleatoric uncertainty may arise due to the absence of variables that capture the impact of anthropogenic activities. Since we utilized variables from Land Surface Models (LSMs) and Climate Indices as inputs to the ML models for reconstructing BNML_TWSA, the influence of anthropogenic activities is not represented by these variables adequately.

[Figure]

Figure 1: Spatial distribution and bar plot of selected predictors using Bayesian network. P, P_1, and P_2 represent precipitation for the current month, one month prior, and two months prior, respectively. Similarly, T, NTWSA, and CTWSA, along with their observations one month prior and two months prior, are used as potential predictors. T denotes temperature, while TWSA from NOAH and the Catchment Land Surface Model (CLSM) are denoted by NTWSA and CTWSA, respectively.

In this study, a model uncertainty assessment is performed for the reconstructed dataset during the model training phase, using the GRACE observations. The uncertainty of the model predictions is quantified by calculating confidence intervals (CIs) of the TWSA estimates. The CI is defined as the point estimate $\pm$ the margin of error, where the margin of error is determined by the product of a confidence coefficient ($C_{confidence}$), derived from the standard normal curve, and the standard error of the point estimate. The standard error of the point estimate is computed using the residuals

from the training set employed in the ML model. The residuals ($\varepsilon$) are calculated as the difference between GRACE JPL Mascon and the reconstructed BNML_TWSA during the training period, as outlined below:

$$GRACE_t = BNML\_TWSA_t + \varepsilon \tag{1}$$

These residuals capture errors arising from data noise and structural model inaccuracies, as discussed earlier. A classical approach to determining the standard error ($\sigma_\varepsilon$) of the residuals is given by:

$$\sigma_\varepsilon = \sqrt{variance(\varepsilon)} \tag{2}$$

For most grid cells, the residuals follow a normal distribution (Fig. 2a). The normality of the residuals was verified using the Shapiro-Wilk test, with normality assumed when the p-value exceeds 0.05. Consequently, it is appropriate to use the standard error to estimate the confidence interval (Humphrey and Gudmundsson, 2019). The confidence interval is calculated as:

$$95\% \ CI = Point \ estimate \pm C_{confidence} \times \sigma_\varepsilon \tag{3}$$

The spatial distribution of the standard error ($\sigma_\varepsilon$) is shown in Fig. 2b. The $\sigma_\varepsilon$ values for grid cells in arid regions are significantly smaller compared to those in other regions, indicating improved accuracy in arid areas. This observation aligns with the findings of Humphrey and Gudmundsson (2019).

[Figure]

Figure 2: Characteristics of residuals of reconstructed BNML_TWSA computed against GRACE JPL Mascon during training period. a) Shapiro-Wilk normality test result on residuals and b) Standard error of residuals.

Climate change and anthropogenic activities are critical factors that can introduce additional uncertainties into the assessment of terrestrial water storage. These uncertainties arise from factors such as land-use changes, irrigation practices, and urbanization, which significantly influence regional water storage dynamics. In this study, variables derived from LSMs were utilized as potential predictors. However, future research could benefit from incorporating input variables from GHMs to better account for anthropogenic influences. GHMs are particularly well-suited for modeling human interventions in water resources, offering a more realistic representation of these activities (Bibi et al., 2024). It is important to acknowledge that both LSMs and GHMs have inherent limitations when utilized as physically-based sources of TWSA (Bibi et al., 2024). The integration of machine learning (ML) models with physical models can help address these limitations, reducing errors in hydrological analyses (Xu et al., 2014). Numerous studies have demonstrated that ML models frequently outperform

traditional hydrological models in various applications (Kim and Kim, 2021; Liang et al., 2023). This suggests that leveraging ML models, alongside advancements in physical modeling, holds great promise for improving the accuracy and reliability of hydrological assessments.

**RC 1.10:** The structure of the published dataset does not follow any data standards. Further the naming of the downloadable zip file, Mississipi_Data.zip does not comply with the contained global grids. You should use descriptive filenames and use modern standard data formats, e.g., CDF conform (netCDF) self-describing data, geotiff, ... and a self describing tree structure. You can get inspiration for instance from other publications in ESSD

**Response:** The reconstructed BNML_TWSA dataset has now been published in a standard netCDF (.nc) format, specifically covering all grid cells within the Mississippi River Basin. This format was chosen for its wide acceptance and compatibility with various data analysis and visualization tools, ensuring ease of use for researchers and practitioners. Following the acceptance of this manuscript, we plan to expand the dataset's availability by uploading a comprehensive file containing reconstructed BNML_TWSA data for all grid cells across the globe.

**Minor things**

**RC 1.11:** L86/87: no commas in large numbers.

**Response:** Commas are added to improve the readability of large numbers, and similar occurrences are checked and corrected in other places.

**RC 1.12:** Table 1: Provide not only publications for the data-sets but also the DOI references where they can be obtained; add the acronyms / abbreviations that you later use in the analysis and figures (e.g. Fig. 2, NTWSA, CTWSA)

**Response:** We have added the DOI references of data sources. Additionally acronyms / abbreviations are added for better understanding.

**RC 1.13:** L199: you name three types of ML algorithms but then are 4 listed and described

**Response:** We have corrected the sentence in the revised manuscript (P:10, L:220-222) as follows: "In this study, four types of machine learning algorithms have been used: neural network-based (CNN), kernel-based (SVR), tree-based (ETR), and an ensemble of these three (CNN, SVR, and ETR) as stacking ensemble regression (SER)."

**RC 1.14:** L274: That's the third different usage of P in the manuscript (Probability, Precipitation, and Prediction in the evaluation metrics)

**Response:** We thank the reviewer for highlighting this abbreviation mistake in the manuscript. In the revised manuscript, 'P' is used exclusively for Precipitation. We have used 'Pr' to represent probability and 'S' to represent the simulated/reconstructed TWSA, as shown below:

$$BIC = \sum_{i=1}^{N} \log\left(Pr(X_i \,|\, MB(X_i))\right) - \frac{d}{2}\log(N) \tag{4}$$

$$CC = \frac{\sum_{i=1}^{n}(O_i - \bar{O})(S_i - \bar{S})}{\sqrt{\sum_{i=1}^{n}(O_i - \bar{O})^2}\sqrt{\sum_{i=1}^{n}(S_i - \bar{S})^2}}, CC \in [-1, 1] \tag{5}$$

where, $O_i$ and $S_i$ represent the TWSA from GRACE/GRACE-FO and the simulated/reconstructed TWSA, respectively, with $\bar{O}$ and $\bar{S}$ denoting their respective means.

**RC 1.15:** L279: A grid is usually defined as a collection of adjacent pixels. You are using the term grid instead of pixel. I suggest to change it to either pixel or grid cell / cells.

**Response:** We have addressed this issue by ensuring consistent terminology throughout the revised manuscript. Specifically, we used the term "grid cell" across all sections to maintain clarity and uniformity.

**RC 1.16:** Fig.2 Expand acronyms in the figure caption, make the caption more explanatory. From the colors it appears that several optimal predictors overlap for the same regions / pixels

**Response:** We have expanded the figure caption in the revised manuscript, as shown below (P:14): "Spatial distribution and bar plot of selected predictors using Bayesian network. P, P_1, and P_2 represent precipitation for the current month, one month prior, and two months prior, respectively. Similarly, T, NTWSA, and CTWSA, along with their observations one month prior and two months prior, are used as potential predictors. T denotes temperature, while TWSA from NOAH and the Catchment Land Surface Model (CLSM) are denoted by NTWSA and CTWSA, respectively."

**RC 1.17:** Fig.3 Avoid red and green in the same figure (colorblind check, you can use https://www.color-blindness.com/coblis-color-blindness-simulator/ for checking)

**Response:** In the revised manuscript, we have replotted the figures with color-blind-safe color palettes, wherever applicable.

**RC 1.18:** L333: grid-based $->$ pixel based

**Response:** We have revised our manuscript based on the reviewer's comment # RC 1.15. Accordingly, the revised line is modified as shown below (P:20, L:380-381): "The leader model, constructed for each global grid cell, is utilized to generate a GRACE-like TWSA series from April 2002 to December 2022 using the input parameter set selected by the BNs for each grid cell."

**RC 1.19:** Fig. 6: describe gray bars in figure caption (gaps in GRACE solutions)

**Response:** the figure caption is modified in the revised manuscript as follows (P:23)

'Figure 9. Comparison of TWSA time series from April 2002 to December 2022 (GRACE period). Vertical gray bars indicate missing GRACE observations'

**RC 1.20:** Fig. 7: Change 1:1 line to non-dashed gray with thicker linewidth to make it distinguishable from the data. Use colors with better contrast for BNML and CTSWA

**Response:** We have replotted this figure with a non-dashed gray 1:1 line, along with distinguishable colors for BNML_TWSA and CTWSA, as shown in Fig. 10 of the revised manuscript (P:24).

**RC 1.21:** Abstract L18: remove "and updates will be published when needed"

**Response:** We have removed this line from the abstract of the revised manuscript.

**Reviewer 2**

**General comments**

Mandal et al. describes a data-driven reconstructed global product of TWS, namely BNML_TWSA. A Bayesian Network technique is used to find an optimal set of predictors. Then, several ML algorithms are trained at each grid cell. The authors choose the best ML algorithm for each grid cell to fore- and hind-cast TWS across the globe. Compared with several existing reconstructed TWS products and estimates by land surface models, the new product shows better agreements with GRACE observations at grid cell and basin scales, with being capable to capture some historical hydroclimatic extreme events. Based on the evaluation results, the authors conclude that the newly developed TWS product is reliable and can be used for hydroclimatological studies.

Overall, I find the manuscript easy to follow. The topic is relevant to the journal, as TWS is an influential variable in many aspects of the Earth system functioning. The evaluation across space and time done by the authors is informative to see how good the ML-derived product performs. However, I still have some concerns regarding the robustness of the new product, majorly coming from the lack of (source of) uncertainty and the way it is evaluated. Please find the comments below. I hope they are helpful to improve the manuscript.

**Response:** We sincerely thank the reviewer for recognizing the relevance of our manuscript to the scope of the journal. We greatly appreciate the time and effort invested in reviewing our work and providing valuable feedback. In the following responses, we have endeavored to address all the concerns and suggestions raised by the reviewer comprehensively. Where applicable, we have revised the manuscript to incorporate the recommended changes, ensuring that the final version aligns with the journal's standards and expectations.

**Major comments**

**RC 2.1:** The the (potential) source(s) of uncertainty needs to be discussed, which the current version of manuscript lacks. There could be some potential sources of uncertainty. First, I wonder if there is any overfit issue in the product as it is fully based on ML and the learning has been done for each grid cell in the study domain. In addition, I wonder if there is any way for BNML_TWSA to capture human impact on TWS and its trend. If no, then it can either be regarded as a source of uncertainty and be specified or be precluded from the model training. If yes, then the authors may add explanations. Lastly, as BNML_TWSA highly depends on TWSA estimates by selected LSMs, the common errors by LSMs, such as the phase shift in mean seasonal cycle (e.g., Bibi et al., 2024) or worse performance in (semi-) arid regions, could propagate to the results. The performance of BNML_TWSA can also potentially be influenced by the precipitation product used.

Bibi, S., Zhu, T., Rateb, A., Scanlon, B. R., Kamran, M. A., Elnashar, A., Bennour, A., and Li, C.: Benchmarking multi-model terrestrial water storage seasonal cycle against Gravity Recovery and Climate Experiment (GRACE) observations over major global river basins, Hydrol. Earth Syst. Sci., 28, 1725–1750, https://doi.org/10.5194/hess-28-1725-2024, 2024.

**Response:** We have incorporated a new subsection into the manuscript dedicated to the evaluation of uncertainties, limitations and the discussion of the future scope of this study (P:38-40, L:520-572).This subsection is also presented in response to Reviewer Comment # RC 1.9 within this document.

Regarding the issue of overfitting, the reconstructed TWSA is evaluated during the testing period against GRACE TWSA to check and prevent any overfitting issues. Additionally, the optimal predictor selection approach adopted in this study helps in minimizing the chances of overfitting. This information has been incorporated into the introduction section as follows (P:3 ,L:89-92)

"Selection of optimal predictors is not just a methodological novelty, but a critical step to ensure that the model prioritizes the most relevant and physically meaningful predictors. This approach reduces noise, minimizes overfitting, and enhances interpretability, making the final product more scientifically robust and practically useful (Das and Chanda, 2024)."

.

**RC 2.2:** I think that readers or potential data users can benefit from additional evaluations. It can be seen that the better performance of BNML_TWSA is expected, as 1) it uses TWSA from LSM(s) as a predictor for many grid cells, 2) the authors choose the best ML algorithms among trained and tested for each grid, and 3) the model is evaluated using TWSA time series. For example, the GRACE-REC by Humphrey and Gudmundsson (2019) that the authors used in the comparison is calibrated against the detrended and deseasonalized TWSA time series, which therefore may not be as good as BNML_TWSA by the design. The results from the evaluation can be seen as the strength of BNML_TWSA, but BNML_TWSA can also benefit from being fairly evaluated against variables that it is not trained with. Evaluating BNML_TWSA using independent variables is important to prove it's ability to extrapolate, because the product has already learned partly the GRACE TWSA information via CTWSA which assimilates GRACE TWSA observations (this fact should have been noted in the main text, I think). As a possible, but not nessesarily the only, way for the additional evaluation, one can suggest that the authors can repeat the evaluation done by Humphrey and Gudmundsson (2019). In the paper, GRACE-REC is evaluated with several independent datasets including sea level budget, streamflow measurements, and basin-scale water balance. This repetition can also work well to compare BNML_TWSA with GRACE-REC. Another way can be evaluating BNML_TWSA at seasonal and interannual (i.e., detrended and deseasonalized) temporal scales. What makes the evaluation at these temporal scales good is because the temporal scales are of a strong interest in multiple communities (e.g., carbon cycle, hydrology, and climate), so the results can be informative for both the product itself and potential users; also it can be regarded as a more fair way to examine BNML_TWSA as it is not trained at this temporal scales.

**Response:** We appreciate the reviewer's valuable suggestion regarding additional evaluations of BNML_TWSA. In the revised manscript, we have included a new data evaluation section, which is also presented below.

**Comparison with streamflow measurements based on basin-scale water balance**

TWS change can be used to estimate streamflow measurement based on the water balance equation for moderately large ($> 100,000$ km$^2$) river basins (Humphrey and Gudmundsson, 2019). The streamflow (Q) based on the water balance model over a watershed may be expressed as:

$$Q = P - ET - \Delta S \tag{6}$$

where the water balance components P and ET are precipitation and evapotranspiration respectively, and $\Delta S$ denotes the TWS change over a time step. Comparison of evaluated streamflow using BNML_TWSA, GRACE TWSA, TWSA from Humphrey and Gudmundsson (2019) (JPL_MSWEP and JPL_ERA5), and Sun et al. (2020) (DNN_JPL-M and DNN_CSR-M) is discussed in this section. Out of the eleven river basins considered in this study, six basins—one from each continent—were selected based on the availability of streamflow data. More details of the six selected basins and their streamflow observation stations are depicted in Table 1. Streamflow observations are predominantly acquired from the Global Runoff Data Centre (GRDC), except for the Godavari River in India, where the streamflow data is sourced from the Central Water Commission (CWC).

Table 1: Details of Basins and Streamflow Observation Locations for Six Global River Basins. Sources: Global Runoff Data Centre (GRDC; `https://portal.grdc.bafg.de`) and Central Water Commission (CWC; `https://indiawris.gov.in`), India

| River Basin | Source | Station for streamflow Observation | Period of streamflow Observation | Drainage Area (km$^2$) |
|---|---|---|---|---|
| Amazon | GRDC | Obidos | April 2002 - December 2019 | 4,671,462 |
| Danube | GRDC | Ceatal Izmail | April 2002 - December 2010 | 779,812 |
| Godavari | CWC | Polavaram | January 2003 - December 2020 | 312,812 |
| Mississippi | GRDC | Vicksburg | April 2002 - October 2022 | 2,918,820 |
| Murray-Darling | GRDC | Lock 1 Downstream | April 2002 - June 2023 | 770,171 |
| Zambezi | GRDC | Katima Mulilo | April 2002 - July 2021 | 334,883 |

Observations of terrestrial water balance components for large river basins worldwide are limited, with sparsely distributed gauges for precipitation and even fewer observations for evapotranspiration. However, due to the availability of data from satellite sensors and outputs from global land surface models, it is possible to analyze the water balance of river basins with sparse observations. Details of the collected dataset and sources are presented in Table 2. Precipitation (P) data from five different sources were collected for each grid cell within these river basins. The basin-scale average of all five precipitation products (GLDAS, GPCC, GPCP, IMERG, and PERSIANN) is considered as the 'observed' precipitation for that particular basin. Similarly, for evapotranspiration (ET), the average of three products (GLDAS, FLDAS, and GLEAM) is considered the 'observed' ET for that basin. In this study, $\Delta S$ for $t^{\text{th}}$ month is calculated as the central difference of terrestrial water storage anomalies, as shown below.

$$\Delta S = \frac{(TWSA_{t+1} - TWSA_{t-1})}{2} \tag{7}$$

Table 2: Overview of Global Precipitation, Evapotranspiration and Storage change Data Products Utilized for Streamflow Calculations

| Dataset | Spatial Resolution | Temporal Resolution | Reference and data source |
|---|---|---|---|
| **Precipitation (P)** | | | |
| GLDAS | 0.25° | 1 month | Rodell et al. (2004) `https://doi.org/10.5067/SXAVCZFAQLNO` |
| GPCC | 0.25° | 1 month | Schneider et al. (2008) `https://dx.doi.org/10.5676/DWD_GPCC/CLIM_M_V2022_025` |
| GPCP | 0.5° | 1 month | `https://doi.org/10.5067/MEASURES/GPCP/DATA304` |
| IMERG | 0.1° | 1 month | `https://doi.org/10.5067/GPM/IMERG/3B-MONTH/07` |
| PERSIANN | 0.25° | 1 month | Ashouri et al. (2015) `https://www.ncei.noaa.gov/data/precipitation-persiann/access/` |
| **Evapotranspiration (ET)** | | | |
| GLDAS | 0.25° | 1 month | Rodell et al. (2004) `https://doi.org/10.5067/SXAVCZFAQLNO` |
| FLDAS | 0.1° | 1 month | `https://doi.org/10.5067/5NHC22T9375G` |
| GLEAM | 0.25° | 1 month | Martens et al. (2017); Miralles et al. (2011) `https://www.gleam.eu` |
| **Storage change (ΔS)** | | | |
| GRACE (JPL mascon) | 0.5° | 1 month | Watkins et al. (2015) `https://doi.org/10.5067/TEMSC-3JC63` |
| BNML_TWSA | 0.5° | 1 month | Mandal et al. (2024) `https://doi.org/10.6084/m9.figshare.25376695` |
| JPL_MSWEP | 0.5° | 1 month | Humphrey and Gudmundsson (2019) |
| JPL_ERA5 | 0.5° | 1 month | Humphrey and Gudmundsson (2019) |
| DNN_JPL-M | 1° | 1 month | Sun et al. (2020) |
| DNN_CSR-M | 1° | 1 month | Sun et al. (2020) |

Using the water balance components described in the previous section, the streamflow for each basin is calculated using various TWSA products, including GRACE, BNML_TWSA, JPL_MSWEP, JPL_ERA5, DNN_JPL-M, and DNN_CSR-M. This computation is performed based on the terrestrial water balance equation (Eqn. 6). The computed Q values are compared with the observed Q values from the station, and the corresponding correlation coefficients (CC) are determined. Figure 3 presents the correlation coefficient (CC) values as a heatmap for all six river basins, highlighting the performance of BNML_TWSA. At the Amazon basin, BNML_TWSA demonstrates strong performance with a CC of 0.89, comparable to GRACE and JPL_MSWEP (CC: 0.9) and JPL_ERA5 (CC: 0.89). In the Danube and Godavari basins, BNML_TWSA outperforms all other TWSA products, achieving the highest CC values, although other products also perform well. For the Mississippi basin, BNML_TWSA, along with DNN_JPL-M and DNN_CSR-M, achieves the highest CC value of 0.7. At the Murray-Darling basin, all TWSA products show minimal CC values due to the negligible magnitude of observed streamflow at the basin outlet. At the Zambezi basin, JPL_MSWEP performs best with a CC of 0.46, whereas BNML_TWSA achieves a CC value of 0.35. This evaluation highlights the superior and/or comparable performance of BNML_TWSA across most basins. The timeseries of the streamflow computed using BNML_TWSA ($Q_{BNML\_TWSA}$) is presented alongside the observed streamflow ($Q_{Observed}$) and the streamflow computed using GRACE TWSA ($Q_{GRACE}$) in Fig. 4. The time series plot (Fig. 4) clearly demonstrates that $Q_{BNML\_TWSA}$ aligns more closely with $Q_{Observed}$ compared to $Q_{GRACE}$. For the Murray-Darling River Basin, the magnitude of $Q_{Observed}$ is negligible due to the large amount of water withdrawal for irrigation and consumption, in addition to heavy regulation (Candogan Yossef et al., 2012). All TWSA products struggle to capture the pattern of low streamflow in the Murray-Darling River Basin.

[Figure]

Figure 3: Basin wise CC values obtained against observed Q and computed Q from water balance using TWSA data from GRACE, BNML_TWSA and other studies.

[Figure]

Figure 4: Comparison of observed streamflow ($Q_{Observed}$), Q obtained from water balance using GRACE TWS data ($Q_{GRACE}$) and Q obtained from water balance using BNML_TWSA TWS data ($Q_{BNML\_TWSA}$).

**RC 2.3:** I think that the title is misleading. What is the role of the feature selection (i.e., BN) on BNML_TWSA? The title can be seen that using the optimal set of feature given by BN is the key to improve the ML based product in the study, but the relevant section or explanation cannot be seen from the current version of manuscript. So, the contribution of deploying the BN technique to the quality of BNML_TWSA could be more elaborated, or the title could be updated.

**Response:** We appreciate the reviewer's concern and have accordingly updated the title of our manuscript in the revised version.

**RC 2.4:** What is the value of examing multiple algorithms for each grid cell? It's clearly reported in the manuscript (e.g., Figure 3) that the spatial pattern leader model is very heterogeneous. However, it has not been reported how different the performance of tested models are, and what the differences are in the actual estimates. I wonder the actual influence on the resulted time series would be minor (e.g., a comparison between the current BNML_TWSA and another BNML_TWSA using the algorithm with the poorest performance for each grid cell), as many ML models usually show similarly good performance.

**Response:** A noticeable improvement is observed across most grid cells globally when the best machine learning (ML) model is selected for each grid cell from the four different models trained for that specific grid cell. This model selection process also aids in reducing uncertainties arising from model inadequacies and/or gaps in knowledge. The details of this selection process have been included in the revised manuscript under the **"Grid-Specific Leader Models"** section, as outlined below.

**Grid specific leader models**

The predictors selected by the BNs in each grid are used as input to predict the TWSA using the four ML algorithms mentioned earlier: CNN, SVR, ETR, and SER. The grid wise leader ML algorithm is identified based on the Pearson correlation coefficient (CC) between predicted TWSA and GRACE TWSA for the test period. The performance difference between the leading ML algorithm and the worst performing ML model is depicted in Fig. 5. Although the improvement in terms of CC value difference is not large for all grid cells globally, more than 15.5% of grid cells show improvements greater than 0.2, while an additional 16.5% of grid cells exhibit improvements between 0.1 and 0.2. For the six grid locations demonstrating maximum improvement, the time series and scatter plot is illustrated in Fig. 6. The estimated TWSA by the best-performing model is in good agreement with the observed TWSA during the testing period. This justifies the use of the best-performing (leading model) to predict the TWSA. Fig. 7 depicts the spatial distribution of the leader algorithms over the globe along with frequency as bar plot. ETR performs the best for the maximum number of grids, with a total of 25703, followed by SVR, SER, and CNN, which perform best for 11609, 11069, and 9646 grids respectively. Thus, for most of the river basins including Krishna and Godavari in India, Danube in Europe, Nile, Zambezi and Limpopo in the African continent, Mississippi in the USA and the transboundary GBM and Indus, ETR emerges as the leader model in maximum grids. The contribution of the leader algorithm as a percentage of the total grid points for each river basin is shown in Fig. 7c. It is observed that in the Limpopo river basin, ETR performs best in 89.0% of the grid points, whereas CNN does not perform best in any of the grids in this basin. In the Murray-Darling river basin in Australia, the four ML algorithms show the best performance at approximately equal number of grid points (CNN: 25.9%, SVR: 21.4%, ETR: 26.1% and SER: 26.6%).

[Figure]

Figure 5: Difference between the correlation coefficient (CC) values obtained from the leader ML model and the worst performing ML algorithm in each grid cell for the test period.

[Figure]

Figure 6: Time series (left columns) for six grid cells showing the maximum improvement globally, including observed TWSA and TWSA predicted by the best and worst models. Scatter plots (right columns) compare the TWSA predicted by the best and worst models against the observed TWSA.

[Figure]

Figure 7: a) Frequency, b) spatial distribution of leader machine learning algorithms and c) leader machine learning algorithms in terms of percentage for different river basins

**RC 2.5:** The results and discussion section is mostly about presenting how good the performance metrics for BNML_TWSA is, which could have been deeper to provide more insight about the product's applicability. Having evaluation from more diverse aspect as in the second bullet point would help with improving this aspect.

**Response:** As suggested by the reviewer in comment # RC 2.2, we have included an evaluation of our reconstructed BNML_TWSA in the revised manuscript.

**RC 2.6:** The dataset provided includes estimates for the Mississippi river basin only, while one would expect estimates for the whole global land grid cells, according to the title and abstract.

**Response:** We appreciate the reviewer's concern and would like to clarify that we plan to provide the dataset for the entire globe upon the manuscript's acceptance for publication. To demonstrate the data structure, we have uploaded

datasets for the Mississippi River Basin in standard netCDF (.nc) format during this review stage.

**Minor and technical comments**

**RC 2.7:** The current manuscript has many in-text narrative citations that are wrongly used, e.g., L311, L334-335, and so on.

**Response:** We agree with the reviewer's observation regarding the issues with multiple in-text narrative citations. These errors have been addressed and corrected in the revised manuscript.

**RC 2.8:** The introduction can be improved by better addressing the motivation to have a new product. Currently, it introduces TWS variable and examples of (ML-based) TWS reconstruction studies. The authors introduce that using a feature selection process can be a novelty of BNML_TWSAm, while testing multiple ML algorithms is important, which can be, but are not necessarily the reason to have a new product. The authors could better present the motivation by showing why having (or lacking) feature selection and multiple ML algorithms are critical for users and their science. Or, showing from which aspect existing reconstructed TWS products are less reliable/robust can better show the motivation. This will also help with having a focused presentation in the results section.

**Response:** We sincerely appreciate the reviewer's insightful comments and suggestions regarding the update to the introduction section. In the revised manuscript, we have included text that addresses the motivation for developing a new product, as well as the benefits of selecting optimal predictors. Please refer to the introduction section of the revised manuscript for further details.

**Specific comments**

**Abstract**

**Introduction**

**RC 2.9:** L23: There can be more references, especially ones done at the global scale.

**Response:** In the revised version, we have incorporated additional references to studies conducted globally, utilizing comprehensive land surface and hydrological models, as illustrated below (P:1-2, L:22-24):

'The fluctuations of TWS in both space and time have been comprehensively simulated by employing physically-based land surface models (LSMs) and global hydrological models (GHMs) (Humphrey et al., 2017; Felfelani et al., 2017; Sun et al., 2021).'

**RC 2.10:** L24: This sentence needs more clarification. Which physical processes are missing? What are the influence on the estimates from which aspect?

**Response:** In the revised manuscript, this concern has been addressed as outlined below: 'The fluctuations of TWS in both space and time have been comprehensively simulated by employing physically-based land surface models (LSMs) and global hydrological models (GHMs) (Humphrey et al., 2017; Felfelani et al., 2017; Sun et al., 2021). These models have significant biases due to inherent uncertainty and the lack of some physical processes, such as the lack of modeling human interventions in water resources within LSMs (Bibi et al., 2024). Furthermore, in snow-dominated basins, LSMs often underestimate peak terrestrial water storage anomalies (TWSA), whereas GHMs tend to overestimate them. Similarly, in temperate, arid, and tropical basins, both model types generally underestimate TWSA peaks (Bibi et al., 2024).'

**RC 2.11:** L29: I think that Mo et al. (2022) is not a proper reference for the sentence. The study is to report a new product, not to examine the human and climatic impact on water cycle.

**Response:** We agree with the reviewer that this reference was added in error. In the revised manuscript, we have removed the reference from the specified line.

**RC 2.12:** L41: The reconstruction by Humphrey et al. (2017) is at the global scale. Only the example application is for the Amazon Basin.

**Response:** We thank the reviewer for highlighting this significant error in our manuscript. In the revised version, we have corrected it as follows (P:2, L:44-46): 'Humphrey et al. (2017) established a statistical data-driven model, between GRACE TWSA using deviations in both temperature and precipitation to recreate TWSA from 1985 to 2015 for the entire globe.'

**RC 2.13:** L32-59: This paragraph is basically list up previous studies wtih a few sentences for each. I wonder if this is the best way of storytelling for readers.

**Response:** Thank you for your valuable feedback. In the revised manuscript, we have condensed this section and incorporated the motivation for developing a new product, as well as the benefits of feature selection, as mentioned in comment RC 2.8. This section now provides an overview of TWSA reconstruction studies and outlines the evolution of algorithms, transitioning from empirical models to statistical models, and subsequently to machine learning models. This framework is intended to offer readers a clear understanding of the progression in this field.

**RC 2.14:** L60: The authors could first list up what the categories are.

**Response:** This concern is addressed in the revised manuscript as follows (P:3, L:66-67): 'The ML models used in hydrological studies so far can be broadly divided into two main categories: single algorithm usage and multiple algorithm usage.'

**RC 2.15:** L77-78: As mentioned above, there need to be more elaboration on the importance of the feature selection on the TWSA reconstruction and the applications.

**Response:** We sincerely appreciate the reviewer's insightful comments and suggestions regarding the motivation for developing the new TWSA product and the importance of feature selection and multiple ML algorithms. In response, we have revised the introduction to better articulate the scientific and practical rationale behind our approach.

Specifically, we have emphasized the role of feature selection in improving the robustness and reliability of the TWSA reconstruction. Feature selection is not just a methodological novelty, but a critical step to ensure that the model prioritizes the most relevant and physically meaningful predictors. This approach reduces noise, minimizes overfitting, and enhances interpretability, making the final product more scientifically robust and practically useful (Das and Chanda, 2024). Additionally, we have elaborated on how existing TWSA products often lack systematic feature selection, leading to potential degradation in their reliability and applicability.

We have also highlighted the importance of testing multiple ML algorithms to ensure methodological robustness and identify the optimal approach for TWSA reconstruction. Different algorithms have varying strengths in handling the non-linearities and complexities inherent in hydrological systems. By comparing multiple approaches, our study offers a comprehensive evaluation that benefits both users and researchers by providing a robust framework for estimating TWSA.

Furthermore, we have addressed the limitations of existing reconstructed TWSA products, such as their sensitivity to predictor selection, lack of uncertainty quantification, and limited applicability across regions. These gaps demonstrate the need for a new approach that integrates feature selection and multiple ML algorithms to produce a more reliable and generalizable product.

These revisions aim to clarify the motivation for the study and ensure a more focused presentation in the results section, as suggested by the reviewer. Thank you for this valuable feedback.

**Data and Processing**

**RC 2.16:** L108: This counters the sentence L78-80

**Response:** The short data gaps (1-2 months) are filled using trained ML models. We have not performed interpolation of intermittent gaps, detrending, deseasoning, or decomposing signals.

**RC 2.17:** L112-113: Are there any reasons to choose Noah and CLSM specifically?

**Response:** Many global and regional studies have utilized the Noah and Catchment Land Surface Model (CLSM) to reconstruct TWSA. For instance, Sun et al. (2019, 2020); Jing et al. (2020); Humphrey et al. (2017) have successfully employed these models in their research.

**RC 2.18:** L125-126: Why doesn't LSMs fully use the information? This can be more elaborated.

**Response:** We have added a reference where the study justifies the inclusion of precipitation and temperature data as input variables, although these were part of LSM forcing:

"Sun, A. Y., Scanlon, B. R., Save, H., & Rateb, A. (2021). Reconstruction of GRACE total water storage through automated machine learning. Water Resources Research, 57, e2020WR028666. https://doi.org/10.1029/2020WR028666"

**RC 2.19:** L129: I think that the time period of analysis hasn't specified before.

**Response:** In the 'Data and processing' section, we have included the period of analysis as follows (P:4, L:112):

"The entire period of analysis spans from 1960 to 2022."

**RC 2.20:** L143: Although CLSM provides TWS directly, the authors should be able to refer to other materials to know which processes CLSM accounts for to calculate TWS. Please add this description, also plase update Table 1 accordingly.

**Response:** We have incorporated a reference to other studies that utilized terrestrial water storage (TWS) data from CLSM. The added reference is as follows:

"Sun, A. Y., Scanlon, B. R., Save, H., & Rateb, A. (2021). Reconstruction of GRACE total water storage through automated machine learning. Water Resources Research, 57, e2020WR028666. https://doi.org/10.1029/2020WR028666"

**RC 2.21:** Eq.1: Sun et al (2019) mentioned that Noah does not account for surface water storage. Please clarify this.

**Response:** The equation used by Sun et al. (2019) is as follows:

$$TWS = SnWS + CWS + SWS + SMS + GWS \tag{8}$$

where SnWS represents snow water storage, CWS is canopy water storage, SWS is surface water storage, SMS is soil moisture storage, and GWS is groundwater storage.

In our study, we combined the CWS and SWS and referred to it as canopy and surface water storage (CSWS).

$$TWS = SnWE + SMC + CSWS \tag{9}$$

where SnWE represents snow depth water equivalent, SMC is soil moisture content, and CSWS is canopy and surface water storage.

**RC 2.22:** L149: "may be" or "can be"? It's a bit weird to use "may be" in this sentence.

**Response:** We have corrected this mistake in the revised manuscript.

**RC 2.23:** L152-154: It was not clear for me if the prior months are for P and T or for TWSA, too. I suggest to rephrase the sentence.

**Response:** This line has been revised in the updated manuscript as follows (P: 7, L:170-171):

"$X$ includes CTWSA, NTWSA, P and T for the current month, as well as for one and two months prior. Additionally, it includes three climate indices (DMI, NAO, ONI) for the current month."

**RC 2.24:** L153: Why aren't the prior months used for the climate indices?

**Response:** The Oceanic Niño Index (ONI) is a three-month running mean of SST anomalies from the NOAA Extended Reconstruction Sea Surface Temperature Version 5 (ERSSTv5) dataset in the Niño 3.4 region (5°N-5°S, 120°-170°W). Therefore, it incorporates the sea surface temperatures of both previous and upcoming months to calculate the ONI for the current month. This is why the prior months have not been incorporated for this index. To standardize the number

of climate indices and control the total number of input variables for BN, prior climate indices have not been utilized.

**RC 2.25:** L155: SVR and ETR need to be introduced as their full name.

**Response:** This line is updated in the revised manuscript as follows:

"Four ML algorithms, namely Convolutional Neural Network (CNN), Support Vector Regression (SVR), Extra Trees Regressor (ETR), and Stacking Ensemble Regression (SER), are trained to solve the regression problem described in Eqn. 2"

**RC 2.26:** Figure 1: For the correlation coefficient, CC has been used through out the manuscript, instead of R.

**Response:** In the revised version of the manuscript, we have corrected this mistake in Fig. 1 (P:8).

**RC 2.27:** L231: 'built', not 'build', I think

**Response:** We rectified this mistake in the revised manuscript.

**RC 2.28:** L233: Is the feature selection procudure in ETR independent to BN?

**Response:** Yes, this process is independent to BN and specific to ETR.

**RC 2.29:** L266-267: For each grid cell, does BN give the optimal set of predictors to each ML algorithm or is the set common for all ML algorithms? If it's the former, how can one be sure that different ML algorithms share the same optimal set of predictors?

**Response:** For each grid cell, BN provide a optimal set of predictors, which is common for all ML algorithms.

**RC 2.30:** Eq.5: On the right hand side, does the denominator use Pi-Pbar or Oi-Obar? The typical NSE equation uses observations for the denominator. Please check this.

**Response:** We thank the reviewer for highlighting this mistake in the NSE equation. In the revised version, we have corrected this as shown below. We used 'S' instead of 'P' to represent the simulated/reconstructed TWSA in the revised manuscript.

$$NSE = 1 - \frac{\sum_{i=1}^{n}(O_i - S_i)^2}{\sum_{i=1}^{n}(O_i - \bar{O})^2}, NSE \in (-\infty, 1] \tag{10}$$

**Results and Discussions**

**RC 2.31:** Figure 2: Would it be reasonable to interpret the results as the emerging importance of the variables to the global TWSA? For example, can the results be seen as that North Atlantic Oscillation has the least influence on the collective gridwise TWSA among the three modes of climate variability?

**Response:** The North Atlantic Oscillation (NAO) was selected by the BN for only a minimal number of grid cells globally. This suggests that the NAO has the least influence on gridwise TWSA.

**RC 2.32:** L291: Please see the comment for Figure 1

**Response:** This error has been corrected in the revised manuscript.

**RC 2.33:** Figure 3: Is it expected that ETR would pop up as the leader? Is there any possible explanation for this, based on the nature of each algorithm?

**Response:** It is not anticipated that any single machine learning (ML) algorithm will perform optimally across the majority of grid cells globally, due to the heterogeneous characteristics of each grid cell. These variations include factors such as land use types, climatic conditions, and differences in meteorological and other forcing variables.

**RC 2.34:** L306-307: It is not clear that BNML_TWSA performs better than LSMs, especially for the case of CTWSA. Could be clearer with histogram of metrics or the map of differences in metrics between BNML_TWSA and LSMs.

**Response:** To enhance interpretation, we have included a plot of the cumulative distribution functions (CDFs) for the CC, NSE, and KGE metrics, as shown below:

The cumulative distribution functions (CDFs) of the CC, NSE, and KGE metrics are presented in Figure 8. These CDFs demonstrate that BNML_TWSA exhibits significantly superior performance, characterized by substantially higher CC, NSE, and KGE values (P:18, L:351-353).

[Figure]

Figure 8: Cumulative Distribution Functions (CDFs) of the Correlation Coefficients (CC), Nash-Sutcliffe Efficiency Coefficient (NSE), and Kling-Gupta Efficiency (KGE) values.

**RC 2.35:** Figure 4: It should be mentioned that BNML_TWSA also shows significant biases in cold or arid regions, and even in a wet region (e.g., a part of the Congo Basin).

**Response:** These limitations of BNML_TWSA have been incorporated into the revised version of the manuscript **(P:40, L:559-570)**.

**RC 2.36:** Figure 6: What does the shaded area stand for? What is the rationale that BNML_TWSA captures the GRACE TWSA trend, for example, in Indus and GBM Basins, where the TWSA trend would largely be affected by human activity?

**Response:** The vertical gray bars in this figure represent missing GRACE observations. In the revised manuscript, this information has been incorporated in the figure caption as follows (P:23): "Figure 9. Comparison of TWSA time series from April 2002 to December 2022 (GRACE period). Vertical gray bars indicate missing GRACE observations."

We appreciate the reviewer's observation. The time series plot demonstrates that BNML_TWSA captures the GRACE TWSA trend more effectively than NTWSA and CTWSA for most of the river basins, including the Indus and GBM Basins, throughout the time period.

**RC 2.37:** Section 4.5: It is great to prove the ability of BNML_TWSA to capture the hydrological extreme events that the MLs were not informed with. However, this section is only mentioning a specific type of event, flood. One could compare the historical TWSA time series of several basins in different climate zones with the corresponding time series of climate indices, drought indices, or precipitation.

**Response:** We sincerely appreciate the reviewer's insightful comments and suggestions regarding comparisons with historical TWSA time series and corresponding time series of climate indices and drought indices. It is worth noting that over 100 drought indices have been proposed to date, addressing various types of drought, including meteorological, agricultural, hydrological, and socioeconomic droughts. A detailed investigation is necessary to explore potential teleconnections between drought occurrences and large-scale climate indices. Furthermore, aspects such as whether precipitation or geographic characteristics predominantly control the groundwater response time to drought, among other critical factors, warrant an independent and comprehensive study. These significant dimensions of drought analysis deserve a dedicated spatial investigation to ensure a thorough and focused exploration.

**RC 2.38:** Figure 9: I feel that the way to show the ability of BNML_TWSA to capture the historical flood events can be seen as inappropriate. For example, in the map of the USA, I can see many other grid cells as bluish as ones in the green box. Does it mean that all the grid cells similarly bluish as ones in the green box are flooded?

**Response:** This map illustrates the difference between the monthly and long-term mean monthly BNML_TWSA datasets.

Extreme events, such as floods, are depicted in the figures for a specific month by grid cells shaded in blue, which can also indicate groundwater recharge due to rainfall. Additionally, as observed in subplot c of this figure (showing the USA), numerous blue cells near Lake Michigan can be explained by the accumulation of water from several rivers flowing into the lake.

**RC 2.39:** Section 4.6: It is recommended to compare BNML_TWSA and previous studies using independent data sets (please see the first major comment). Also, it needs to be noted that GRACE-REC is calibrated against detrended and deseasonalized GRACE TWSA.
(may not be good at captureing the local TWSA correctly, if GLDAS is calibrated using TWS? BNML_TWSA largely depends on GLDAS LSMs which cannot be reliable at finer spatial scales than the original GRACE spatial resolution) < −− need to check how GLDAS LSMs simulates TWS

**Response:** We sincerely appreciate the reviewer's insightful suggestions regarding the comparison of BNML_TWSA with previous studies using independent data sets. We have evaluated our reconstructed dataset along with datasets from previous studies against observed streamflow. Please refer to the response to # RC 2.2 in this response document.

We have incorporated the 'rec_ensemble_mean' and 'rec_seasonal_cycle' components of the GRACE-REC product by Humphrey and Gudmundsson (2019), which represent the ensemble mean of the TWS reconstruction (deseasonalized) and the seasonal cycle, respectively, to make it equivalent to BNML_TWSA. Additionally, in relation to the TWS simulation process by GLDAS LSMs, we have provided references of other studies that evaluated TWS for both GLDAS LSMs considered in this study. Please refer to the responses to # RC 2.20 and # RC 2.21 in this response document.

**Conclusions**

**RC 2.40:** L432-434: I think that this point can be mentioned in the main text, possibly with a deeper discussion (i.e., implications of the distribution of selected gridwise predictors for the glbola TWSA and a possible explanation).
**Response:** We sincerely appreciate the reviewer's suggestions and in the revised manuscript, we have included this portion in the main text under the subsection titled 'Selected predictors using BN,' as follows (P:13, L:313-318):
"It is noteworthy that, in addition to TWSA from LSMs, the ONI and DMI have been selected as optimal predictors for a substantial number of grid cells. Meteorological variable such as P and T, along with their observations from previous months, have been selected for fewer grid cells globally by the BN compared to ONI and DMI. This can be attributed to the fact that LSMs already incorporate these meteorological variables as forcing inputs. The inclusion of climate indices as potential predictors for a large number of grid cells can be seen as an effort to represent the climate change scenarios of that specific time period."

**Data**

**RC 2.41:** Please provide the unit and the file naming convention.
**Response:** In this version of dataset we have used files in .nc format following naming convention.

**RC 2.42:** This should be also noted in the main text with the number or portion of the grid cells: "# For very few grids BN is not identifying any predictors or only one predictor. Grids with zero or one optimal predictor identified by BN are trained using all 15 potential predictors ('P', 'T', 'NOAH_TWSA', 'CLSM_TWSA', 'DMI', 'NAO', 'ONI', 'P1', 'T1', 'NOAH_TWSA_1', 'CLSM_TWSA_1', 'P2', 'T2', 'NOAH_TWSA_2', 'CLSM_TWSA_2')." Also, I would exclude these grid cells with zero or one predictor identified from Figure 2, if it makes a significant difference.
**Response:** We appreciate the reviewer's suggestions and in the revised manuscript, we have included this portion in the main text under the subsection titled 'Selected predictors using BN,' as follows:
"In a limited number of grid cells (66), the Bayesian Network (BN) did not select any predictors. For an additional 492 grid cells, the BN selected only one predictor, thereby limiting the application of certain machine learning algorithms to

these grid cells. Consequently, for a total of 558 grid cells, which constitute less than 1% of the grid cells considered in this study, the complete set of 15 predictors has been used as potential predictors."

**References**

Ashouri, H., Hsu, K.-L., Sorooshian, S., Braithwaite, D. K., Knapp, K. R., Cecil, L. D., Nelson, B. R., and Prat, O. P.: PERSIANN-CDR: Daily precipitation climate data record from multisatellite observations for hydrological and climate studies, Bulletin of the American Meteorological Society, 96, 69–83, 2015.

Bibi, S., Zhu, T., Rateb, A., Scanlon, B. R., Kamran, M. A., Elnashar, A., Bennour, A., and Li, C.: Benchmarking multimodel terrestrial water storage seasonal cycle against Gravity Recovery and Climate Experiment (GRACE) observations over major global river basins, Hydrology and Earth System Sciences, 28, 1725–1750, https://doi.org/10.5194/hess-28-1725-2024, 2024.

Boergens, E., Kvas, A., Eicker, A., Dobslaw, H., Schawohl, L., Dahle, C., Murböck, M., and Flechtner, F.: Uncertainties of GRACE-Based Terrestrial Water Storage Anomalies for Arbitrary Averaging Regions, Journal of Geophysical Research: Solid Earth, 127, e2021JB022 081, https://doi.org/https://doi.org/10.1029/2021JB022081, e2021JB022081 2021JB022081, 2022.

Candogan Yossef, N., van Beek, L. P. H., Kwadijk, J. C. J., and Bierkens, M. F. P.: Assessment of the potential forecasting skill of a global hydrological model in reproducing the occurrence of monthly flow extremes, Hydrology and Earth System Sciences, 16, 4233–4246, https://doi.org/10.5194/hess-16-4233-2012, 2012.

Das, P. and Chanda, K.: Selection of optimum GCMs through Bayesian networks for developing improved machine learning based multi-model ensembles of precipitation and temperature, Stochastic Environmental Research and Risk Assessment, https://doi.org/10.1007/s00477-024-02856-3, 2024.

Felfelani, F., Wada, Y., Longuevergne, L., and Pokhrel, Y. N.: Natural and human-induced terrestrial water storage change: A global analysis using hydrological models and GRACE, Journal of Hydrology, 553, 105–118, https://doi.org/https://doi.org/10.1016/j.jhydrol.2017.07.048, 2017.

Gao, S., Hao, W., Fan, Y., Li, F., and Wang, J.: A Multi-Source GRACE Fusion Solution via Uncertainty Quantification of GRACE-Derived Terrestrial Water Storage (TWS) Change, Journal of Geophysical Research: Solid Earth, 128, e2023JB026 908, https://doi.org/https://doi.org/10.1029/2023JB026908, e2023JB026908 2023JB026908, 2023.

Humphrey, V. and Gudmundsson, L.: GRACE-REC: a reconstruction of climate-driven water storage changes over the last century, Earth System Science Data, 11, 1153–1170, https://doi.org/https://doi.org/10.5194/essd-11-1153-2019, 2019.

Humphrey, V., Gudmundsson, L., and Seneviratne, S. I.: A global reconstruction of climate-driven subdecadal water storage variability, Geophysical Research Letters, 44, 2300–2309, https://doi.org/https://doi.org/10.1002/2017GL072564, 2017.

Jing, W., Di, L., Zhao, X., Yao, L., Xia, X., Liu, Y., Yang, J., Li, Y., and Zhou, C.: A data-driven approach to generate past GRACE-like terrestrial water storage solution by calibrating the land surface model simulations, Advances in Water Resources, 143, 103 683, https://doi.org/https://doi.org/10.1016/j.advwatres.2020.103683, 2020.

Kalu, I., Ndehedehe, C. E., Ferreira, V. G., and Kennard, M. J.: Machine learning assessment of hydrological model performance under localized water storage changes through downscaling, Journal of Hydrology, 628, 130 597, https://doi.org/https://doi.org/10.1016/j.jhydrol.2023.130597, 2024.

Kim, C. and Kim, C.-S.: Comparison of the performance of a hydrologic model and a deep learning technique for rainfall-runoff analysis, Tropical Cyclone Research and Review, 10, 215–222, https://doi.org/https://doi.org/10.1016/j.tcrr.2021.12.001, 2021.

Liang, W., Chen, Y., Fang, G., and Kaldybayev, A.: Machine learning method is an alternative for the hydrological model in an alpine catchment in the Tianshan region, Central Asia, Journal of Hydrology: Regional Studies, 49, 101 492, https://doi.org/https://doi.org/10.1016/j.ejrh.2023.101492, 2023.

Mandal, N., Das, P., and Chanda, K.: Optimal feature selection for improved ML based reconstruction of Global Terrestrial Water Storage Anomalies, https://doi.org/10.6084/M9.FIGSHARE.25376695, 2024.

Martens, B., Miralles, D. G., Lievens, H., Van Der Schalie, R., De Jeu, R. A., Fernández-Prieto, D., Beck, H. E., Dorigo, W. A., and Verhoest, N. E.: GLEAM v3: Satellite-based land evaporation and root-zone soil moisture, Geoscientific Model Development, 10, 1903–1925, 2017.

Miralles, D. G., Holmes, T., De Jeu, R., Gash, J., Meesters, A., and Dolman, A.: Global land-surface evaporation estimated from satellite-based observations, Hydrology and Earth System Sciences, 15, 453–469, 2011.

Rodell, M., Houser, P. R., Jambor, U., Gottschalck, J., Mitchell, K., Meng, C.-J., Arsenault, K., Cosgrove, B., Radakovich, J., Bosilovich, M., Entin, J. K., Walker, J. P., Lohmann, D., and Toll, D.: The Global Land Data Assimilation System, Bulletin of the American Meteorological Society, 85, 381 – 394, https://doi.org/https://doi.org/10.1175/BAMS-85-3-381, 2004.

Schneider, U., Fuchs, T., Meyer-Christoffer, A., and Rudolf, B.: Global precipitation analysis products of the GPCC, Global Precipitation Climatology Centre (GPCC), DWD, Internet Publikation, 112, 2008.

Sun, A. Y., Scanlon, B. R., Zhang, Z., Walling, D., Bhanja, S. N., Mukherjee, A., and Zhong, Z.: Combining Physically Based Modeling and Deep Learning for Fusing GRACE Satellite Data: Can We Learn From Mismatch?, Water Resources Research, pp. 1179–1195, https://doi.org/10.1029/2018WR023333, 2019.

Sun, A. Y., Scanlon, B. R., Save, H., and Rateb, A.: Reconstruction of GRACE Total Water Storage Through Automated Machine Learning, Water Resources Research, 57, 1–20, https://doi.org/10.1029/2020WR028666, 2021.

Sun, Z., Long, D., Yang, W., Li, X., and Pan, Y.: Reconstruction of GRACE data on changes in total water storage over the global land surface and 60 basins, Water Resources Research, 56, e2019WR026 250, https://doi.org/https://doi.org/10.1029/2019WR026250, 2020.

Watkins, M. M., Wiese, D. N., Yuan, D.-N., Boening, C., and Landerer, F. W.: Improved methods for observing Earth's time variable mass distribution with GRACE using spherical cap mascons, Journal of Geophysical Research: Solid Earth, 120, 2648–2671, https://doi.org/10.1002/2014JB011547, 2015.

Xu, T., Valocchi, A. J., Choi, J., and Amir, E.: Use of Machine Learning Methods to Reduce Predictive Error of Groundwater Models, Groundwater, 52, 448–460, https://doi.org/https://doi.org/10.1111/gwat.12061, 2014.

---

## Author Response (AR2)

**Authors response to reviews for manuscript number #essd-2024-109**

**Anonymous referee #1**

**RC 1.0:** The authors have satisfactorily revised the manuscript, improved readability, and clarified the raised questions. I have a few more minor remarks that should be addressed before final publication.

**Response:** We thank the reviewer for their positive assessment of the revised manuscript and for acknowledging the improvements in readability and clarity. We sincerely appreciate the reviewer's ongoing participation and are committed to addressing the remaining minor comments to finalize the manuscript.

**RC 1.1:** The findings of the uncertainty assessment (Sec. 4.8) should be shortly reflected in the conclusions as well as in the abstract.

**Response:** We thank the reviewer for this valuable suggestion. In the revised manuscript, the findings of the uncertainty assessment have been included in the conclusion section (P: 43, L:612-614), as follows:

"Furthermore, the uncertainty associated with BNML_TWSA is assessed for each grid cell in the form of standard error $(\sigma_\varepsilon)$. The results showed that the standard error of the BNML_TWSA exhibits a smaller magnitude in grid cells located in arid regions compared to those in other regions."

This information is also included in the abstract section (P:1, L:18-19).

**RC 1.2:** Fig. 3: Do you only consider CCs > 0? Otherwise you may find positive differences also by deteriorating from a negative value toward zero.

**Response:** We thank the reviewer for this suggestion. In the revised manuscript, we have excluded the grid cells that have negative CC values for the leading ML model. Specifically, 1,535 grid cells ( 2.65% of the total grid cells considered in this study) have CC values less than zero, and we have excluded these when calculating the difference. Although the improvement in terms of CC value difference (between the leader model and the worst model) is not large for all grid cells globally, more than 14.4% of grid cells show improvements greater than 0.2, while an additional 15.7% of grid cells exhibit improvements between 0.1 and 0.2 (P:15,L:330-335). The updated figure is also shown below.

[Figure]

Figure 1: Difference between the correlation coefficient (CC) values obtained from the leader ML model (excluding negative CC values) and the worst performing ML algorithm in each grid cell during the test period.

**RC 1.3:** It seems very strange to me that your KGE values are always negative while you get positive NSE values in the range of 0.5 to 0.8. I suggest to reassess the computation of KGE or to provide some discussion why these negative values would be feasible. Possibly, KGE' can be used as an alternative to KGE. Please provide the reference for KGE.

**Response:** We sincerely thank the reviewer for highlighting the unusual combination of negative KGE values alongside positive NSE values (0.5 to 0.8). We agree that this required further explanation, and we appreciate you bringing it to our attention.

In the previous version of the manuscript, we employed a modified Kling-Gupta Efficiency (KGE') as defined below (Kling et al., 2012; Gupta et al., 2009)

$$KGE = 1 - \sqrt{(CC - 1)^2 + (\beta - 1)^2 + (\alpha - 1)^2}, KGE \in (-\infty, 1] \tag{1}$$

where $\beta$ and $\alpha$ represents bias error and variability error respectively.

$$\beta = \text{bias ratio} = \frac{\bar{S}}{\bar{O}} \tag{2}$$

$$\alpha = \text{variability ratio} = \frac{\sigma_S/\bar{S}}{\sigma_O/\bar{O}} \tag{3}$$

In this revised manuscript, we have now calculated the KGE according to the original equation presented by Gupta et al. (2009):

$$KGE = 1 - \sqrt{(CC - 1)^2 + (\frac{\bar{S}}{\bar{O}} - 1)^2 + (\frac{\sigma_S}{\sigma_O} - 1)^2}, KGE \in (-\infty, 1] \tag{4}$$

where $\sigma_O$ and $\sigma_S$ represent the standard deviations of observations and simulations, respectively. This correction has resolved the issue of the negative KGE values.

We have also added the appropriate citation for the original KGE formulation (Gupta et al., 2009) to the revised manuscript (P:12-13). Thank you again for your valuable feedback.

**Additional editorial remarks:**

**RC 1.4:** L7: I suggest to change grid cell-based to "raster-based"

**Response:** We have updated the Abstract in the revised manuscript to reflect this change (P:1, L:7).

**RC 1.5:** Table 3 & 4: Captions should not be capitalized

**Response:** We are thankful to the reviewer for for bringing it our notice. In the revised manuscript, we have rectified these mistakes as follows:

**Caption of Table 3**

"Details of basins and streamflow observation locations for six global river basins. Sources: Global Runoff Data Centre (GRDC; `https://portal.grdc.bafg.de`) and Central Water Commission (CWC; `https://indiawris.gov.in`), India"

**Caption of Table 4**

"Overview of global precipitation, evapotranspiration and storage change data products utilized for streamflow calculations"

**RC 1.6:** Figure 4: Units are missing for the axes.

**Response:** We have updated this figure in the revised manuscript and the updated figures is also presented below.

[Figure]

Figure 2: Time series (left columns) for grid cells showing the maximum improvement within the basins considered in this study, including observed TWSA and TWSA predicted by the best and worst models. Scatter plots (right columns) compare the TWSA predicted by the best and worst models with the observed TWSA.

**RC 1.7:** Figure 8: Y-Axis labels should be put on the left side, next to the numbers.

**Response:** This figure has been updated in the revised manuscript according to the reviewer's suggestion. Please find the revised figure below.

[Figure]

Figure 3: Box plot of CC, NSE, RMSE and KGE values for the grid cells of each basin, excluding the outliers.

**RC 1.8:** Figure 9, 11, 19: Units (mm) are missing for the y-axes.
**Response:** We thank the reviewer for bringing it to our notice. In the revised manuscript, we have rectified these figures and have shown the updated versions below.

[Figure]

Figure 4: Comparison of TWSA time series from April 2002 to December 2022 (GRACE period). Vertical gray bars indicate missing GRACE observations.

[Figure]

Figure 5: BNML_TWSA during the pre-GRACE period (Jan 1960 - Mar 2002)

[Figure]

Figure 6: Comparison of observed streamflow ($Q_{\text{Observed}}$), Q obtained from water balance using GRACE TWS data ($Q_{\text{GRACE}}$) and Q obtained from water balance using BNML_TWSA TWS data ($Q_{\text{BNML\_TWSA}}$). RMSE values (right columns) obtained for $Q_{\text{GRACE}}$ and $Q_{\text{BNML\_TWSA}}$ against $Q_{\text{Observed}}$.

**RC 1.9:** Figure 18: Notation (Label) for CC is missing for the color-scale

**Response:** We appreciate the reviewer's constructive feedback. The label "CC" has now been added to the color scale in Figure 18 in the revised manuscript.

**Data product:**

**RC 1.10:** Once you publish the global data set, I encourage you to provide also the global grids as netcdf files – instead of only catchment based portions. But probably this is your intention anyway. I think it would be good to provide also the Model_Input and BN_Predictors data as netCDF grids. For the BN_Predictors, you could use an integer based classification. With your current method, you will have to generate a huge number of text-files for the global grid. Importing these data into models will be much easier based on netCDF. Please check also the CF compliance of your netCDF files. Lat and lon seem to have issues here so that cdo does not recognize the grid type (latlong).

**Response:** After acceptance of the manuscript, we will upload the netcdf files for BNML_TWSA products, Model Input, and BN Predictors using an integer-based classification. Additionally, we will check the CF compliance of our generated NetCDF files. We sincerely apologize for the issue regarding opening the NetCDF files in CDO. While publishing the global dataset, we will ensure it has CF compliance.

**Anonymous referee #2**

**RC 2.0:** I appreciate efforts by the authors to address my comments substantially. Now I only have some minor comments.
**Response:** Thank you for acknowledging our efforts to address your comments. We appreciate your thorough review and are committed to addressing the remaining minor comments.

**Minor and technical comments:**

**RC 2.1:** L37-46: I suggest to add a topic sentence at the head of the paragraph, like in the next paragraph. Currently, the paragraph is just a list of example studies and the reader should guess what the paragraph is about based on them.
**Response:** We appreciate the suggestion and in the revised manuscript (P:2,L:39-41), we have added the following at the head of the paragraph:
"It is imperative that a reliable, long-term continuous TWSA dataset is valuable for various aspects of the functioning of the Earth system including the assessment of basin-scale water balance and local hydrological extremes. Different studies have employed various techniques to reconstruct long-term TWSA beyond the GRACE period."

**RC 2.2:** L66-87 (A) and L88-109 (B): Each of two paragraphs deals with two topics, so it is less clear what each paragraph wants to talk about. Paragraph A deals with the importance of 1) testing multiple ML algorithms and 2) the optimal feature selection. Paragraph B deals with the importance of the optimal feature selection and debriefs the whole study. I would suggest merge the parts about the optimal feature selection as a new paragraph so that there are three paragraphs about multiple ML algorithms, the optimal feature selection, and debriefing the study.
**Response:** We appreciate the suggestion and have updated the concerned portion of the manuscript in the revised version (P:3-4, L:70-115).

**RC 2.3:** Equation 1: I think Eq. 1 can be missled in the current form and a reader may regard that Noah considers surface water storage. Sun et al. (2019) used the equation to show the TWS calculation in general, not for a specific product or model. However, in this study, Equation 1 is to introduce how TWS is calculated in Noah, which does not calculate SWS. So, I suggest to replace CSWS with CWS.
**Response:** In the revised manuscript, we have incorporated the reviewer's suggestion (P:7, L:167-171). The updated content is also shown below.

$$TWS = SnWE + SMC + CWS \tag{5}$$

where SnWE represents snow depth water equivalent, SMC is soil moisture content, and CWS is canopy water storage. We have also updated Figure 1 and Table 1 in the revised manuscript to incorporate this information.

**RC 2.4:** Figure 4: The six grid cells highlighted are all neighboring to each other. I understand the logic to select the grid cells based on the differences in CC, but using the resulting sample group is less informative for the purpose of the figure, I would say. I suggest to consider selecting other grid cells from other regions, too.
**Response:** We appreciate the reviewer's suggestion to select grid cells from other regions. In the revised manuscript, we have updated this figure to include a grid cell showing the maximum improvement within a river basin for each of the 11 river basins considered in this study (P: 15, L:335-338). The updated figure is also provided in Figure 2 of this response document.

**RC 2.5:** L358-360: Two last sentences are duplicated in terms of the message.
**Response:** We sincerely appreciate the reviewer's effort in highlighting these duplicate sentences. In the revised manuscript (P:19, L:364-365), we have corrected this issue as follows: "The results indicate the superior performance of BNML_TWSA compared to the other methods, which is evident from these plots on a global scale."

**RC 2.6:** L417-418: I think it's wrong to say like "the proposed model accurately reproduced the TWSA series during the pre-GRACE period" by comparing three model simulations.

**Response:** In the revised manuscript, we have updated this sentence as follows (P:26, L:423-424): 'The proposed model simulated the TWSA series during the pre-GRACE period and identified recorded climate extreme events that occurred in the hindcast period.'

**RC 2.7:** L435-436: I suggest to specify the period.

**Response:** We sincerely appreciate the reviewer's suggestion. In the revised manuscript (P:29, L:441-442 and P:31, L:449-450), we have specified the period of comparison as follows:

P:29, L:441-442: "Specifically, the period spans from April 2002 to July 2019 for JPL_ERA5 and from April 2002 to December 2016 for JPL_MSWEP."

P:31, L:449-450: "...for the period from April 2002 to July 2018..."

**RC 2.8:** L515-516: It's hard to agree. Both BNML_TWSA and GRACE struggle to get the observation in rather drier basins, and they behave similarly in general in other basins. One can at least highlight the comparable behavior of $Q_{BNML\_TWSA}$, or there would need to be additional information (e.g., metrics) to say that $Q_{BNML\_TWSA}$ performs better clearly.

**Response:** We appreciate the reviewer's observation regarding the need for proper justification of the improved performance of $Q_{BNML\_TWSA}$. In the revised manuscript, we have added RMSE values obtained against observed streamflow for both $Q_{BNML\_TWSA}$ and $Q_{GRACE}$ and depicted them alongside the time series plot (Figure 6 of this response document). Additionally, we have updated this result in the "**Comparison with streamflow measurements based on basin-scale water balance**" sub-section of the revised manuscript (P:36, L:522-524).

**RC 2.9:** L521-524: Is it the uncertainty of GRACE as an assumed ground truth that would afffect the product evaluation, or is the processing errors in GRACE liked to the uncertainty of BNML_TWSA? If the former, it would be better to specify it. If it is the latter, BNML_TWSA is still subject to uncertainties of mascon solutions, although they perform better compared to the spherical harmonic solutions. I would indicate both aspects.

**Response:** We appreciate the reviewer's valuable suggestion. In the revised manuscript, we have mentioned both aspects as follows (P:39, L:529-536):

"There are various sources that contribute to the uncertainties in reconstructed TWSA. The primary source of uncertainties arises from the measurement errors, inherent processing errors, leakage errors and model assumptions associated with the original GRACE data, as documented by Boergens et al. (2022) and Gao et al. (2023). Nevertheless, this issue is effectively mitigated by utilizing the mascon solution, which demonstrates clear superiority over the spherical harmonics data (Kalu et al., 2024). The JPL mascon solution employs a Coastline Resolution Improvement (CRI) filter to minimize leakage errors across land/ocean boundaries. Additionally, gain factors are utilized to further mitigate these leakage errors. Moreover, a Bayesian framework is implemented to more effectively eliminate correlated errors compared to traditional empirical filters (Wiese et al., 2016)."

**RC 2.10:** L526-527: The improvement by multiple MLs is majorly for (semi-)arid regions (Figure 3). This means that, every ML algorithm perform greatly for wetter regions or none of them could not improve the simulation there. So, for wetter regions, I suspect there are still rooms to improve (e.g., the KGE map in Figure 6).

**Response:** We appreciate the reviewer's observation regarding the lower KGE values observed in river basins within wet climatic zones, such as the Amazon and Godavari. We concur that this highlights the potential for improvement. As the reviewer suggests, and as we also note in the manuscript (P: 41-42, L: 576-581), integrating machine learning (ML) models with physical models offers a promising avenue for achieving such improvements.

**RC 2.11:** L556-557: The sigma in arid regions may be small by nature because of the small variance of TWS there. However, I agree that testing multiple ML algorithms partly contributed to the improvement in performance (and small sigma values) in arid regions, referring to Figure 3.

**Response:** We sincerely thank the reviewer for this insightful observation regarding the improvement in performance in arid regions. Indeed, testing multiple ML algorithms contributes to performance enhancement globally. However, this improvement is particularly significant in arid regions.

**RC 2.12:** Please make sure that the uncertainty data from Figure 20b is included when the published datasets are updated.

**Response:** We will upload the uncertainty data in terms of standard error as a netCDF file when the published datasets are updated. This information is also included in the Data Availability section of the revised manuscript.

**References**

Boergens, E., Kvas, A., Eicker, A., Dobslaw, H., Schawohl, L., Dahle, C., Murböck, M., and Flechtner, F.: Uncertainties of GRACE-Based Terrestrial Water Storage Anomalies for Arbitrary Averaging Regions, Journal of Geophysical Research: Solid Earth, 127, e2021JB022 081, https://doi.org/https://doi.org/10.1029/2021JB022081, e2021JB022081 2021JB022081, 2022.

Gao, S., Hao, W., Fan, Y., Li, F., and Wang, J.: A Multi-Source GRACE Fusion Solution via Uncertainty Quantification of GRACE-Derived Terrestrial Water Storage (TWS) Change, Journal of Geophysical Research: Solid Earth, 128, e2023JB026 908, https://doi.org/https://doi.org/10.1029/2023JB026908, e2023JB026908 2023JB026908, 2023.

Gupta, H. V., Kling, H., Yilmaz, K. K., and Martinez, G. F.: Decomposition of the mean squared error and NSE performance criteria: Implications for improving hydrological modelling, Journal of Hydrology, 377, 80–91, https://doi.org/https://doi.org/10.1016/j.jhydrol.2009.08.003, 2009.

Kalu, I., Ndehedehe, C. E., Ferreira, V. G., and Kennard, M. J.: Machine learning assessment of hydrological model performance under localized water storage changes through downscaling, Journal of Hydrology, 628, 130 597, https://doi.org/https://doi.org/10.1016/j.jhydrol.2023.130597, 2024.

Kling, H., Fuchs, M., and Paulin, M.: Runoff conditions in the upper Danube basin under an ensemble of climate change scenarios, Journal of Hydrology, 424-425, 264–277, https://doi.org/https://doi.org/10.1016/j.jhydrol.2012.01.011, 2012.

Wiese, D. N., Landerer, F. W., and Watkins, M. M.: Quantifying and reducing leakage errors in the JPL RL05M GRACE mascon solution, Water Resources Research, 52, 7490–7502, https://doi.org/https://doi.org/10.1002/2016WR019344, 2016.

---

## Author Response (AR3)

**Authors response to reviews for manuscript number #essd-2024-109**

**Topic editor:**

Many thanks for the revision, I will accept the manuscript for final publication after you have published the "netcdf files for BNML TWSA products, Model Input, and BN Predictors using an integer-based classification" and finished the checks for CL compatibility. Also please make sure that you update the data availablity statement accordingly. This includes an update of the DOI for the data if required.

**Response:** We sincerely appreciate the editor's constructive comments and thoughtful review of the manuscript. In response to the editor's suggestions:

1. We have published the NetCDF files for the BNML_TWSA dataset, model inputs, BN Predictors (using an integer-based classification), and the uncertainty associated with the BNML_TWSA in terms of standard error for each grid cell considered in this study. A README file with the data description is also published.

2. The Climate and Forecast (CF) compatibility check has been conducted before publishing the NetCDF files.

3. The Python codes used to run the machine learning models are made available through the same DOI as the datasets.

We are grateful for the editor's guidance.